# The AKT isoforms 1 and 2 drive B cell fate decisions during the germinal center response

Zilu Zhu[1,2], Ashima Shukla[1,2], Parham Ramezani-Rad[1,2], John R Apgar[1,2], Robert C Rickert[1,2]

The PI3K pathway is integral for the germinal center (GC) response. However, the contribution of protein kinase B (AKT) as a PI3K effector in GC B cells remains unknown. Here, we show that mice lacking the AKT1 and AKT2 isoforms in B cells failed to form GCs, which undermined affinity maturation and antibody production in response to immunization. Upon B-cell receptor stimulation, AKT1/2–deficient B cells showed poor survival, reduced proliferation, and impaired mitochondrial and metabolic fitness, which collectively halted GC development. By comparison, *Foxo1*[T24A] mutant, which cannot be inactivated by AKT1/2 phosphorylation and is sequestered in the nucleus, significantly enhanced antibody class switch recombination via induction of activation-induced cytidine deaminase (AID) expression. By contrast, repression of FOXO1 activity by AKT1/2 promoted IRF4-driven plasma cell differentiation. Last, we show that T-cell help via CD40, but not enforced expression of Bcl2, rescued the defective GC response in AKT1/2–deficient animals by restoring proliferative expansion and energy production. Overall, our study provides mechanistic insights into the key role of AKT and downstream pathways on B cell fate decisions during the GC response.

## Introduction

Germinal centers (GCs) are specialized microenvironments within the secondary lymphoid tissues where antigen-activated B cells undergo clonal expansion, immunoglobulin class switching, and affinity maturation. The GC is characteristically polarized into the dark and light zone (DZ and LZ). GC B cells iteratively migrate between the DZ and LZ, undergoing clonal expansion and somatic hypermutation (SHM) in the DZ followed by B-cell receptor (BCR) affinity-based selection in the LZ. GC B cells expressing high-affinity BCRs are positively selected in response to signals provided by cognate T follicular helper (Tfh) cells in the LZ. This process of affinity maturation drives the affinity of serum antibody over time (Rajewsky, 1996; Allen et al, 2007; Victora & Nussenzweig, 2012; Mesin

et al, 2016Rajewsky, 1996; Allen et al, 2007Victora & Nussenzweig, 2012Mesin et al, 2016). There are two models to explain the positive selection mechanism: BCR signaling–based or T-cell help–based selection (Allen et al, 2007; Shlomchik & Weisel, 2012). Recent studies found that BCR signaling to be critical for efficient selection in the LZ because it synergizes with T cell help to enhance the GC and plasmablast response (Turner et al, 2018) and induce expression of c-Myc, a critical driver of B cell proliferation (Luo et al, 2018). These findings indicate that GC B cells can integrate variable inputs from T cells and the BCR in vivo for an optimal process of selection and differentiation. Understanding the intracellular signaling pathways that effect these distinct differentiation pathways is an ongoing challenge.

There is accumulating evidence showing that the PI3K signaling pathway is important in GC B cell selection, stemming from early reports on CD19 function and the role of PI3K subunits and its primary lipid product, PtdIns(3,4,5)P$_3$ (Baracho et al, 2011). Recruitment of the PtdIns(3,4,5)P$_3$–binding Bruton's tyrosine kinase, PDK1, and protein kinase B (AKT) may represent the key downstream events of PI3K in B cells. Indeed, we have demonstrated that PDK1 is essential for peripheral B cell activation (Baracho et al, 2014). PDK1 can phosphorylate AKT, enabling activity to modulate B cell fate and metabolism via the mechanistic target of rapamycin (mTOR) complexes, forkhead box protein 1 (FOXO1), glycogen synthase kinase 3 (GSK3), and the tuberous sclerosis complex. Previous study found that the phosphorylation of S6 and associated mTOR activation, induced by T-cell help in LZ, triggered an anabolic program in B cells that sustained subsequent proliferation in the DZ (Ersching et al, 2017). Two transcription factors, FOXO1 and c-Myc, have been shown to regulate the positive selection process. FOXO1 deficiency abrogates the DZ and results in defective affinity maturation and class switching (Dengler et al, 2008; Dominguez-Sola et al, 2015; Sander et al, 2015). Although c-Myc is expressed in only a small fraction of LZ B cells (centrocytes) in mature GCs, its activity is pivotal for GC initiation, maintenance, and positive selection (Calado et al, 2012; Dominguez-Sola et al, 2012). Previously, we showed that GSK3 is required for the generation and maintenance of GC B cells, functioning as a metabolic checkpoint regulator in part by repressing c-Myc–dependent growth (Jellusova et al, 2017). Impaired affinity maturation is also found in mTOR gain of function

[1]Tumor Microenvironment and Cancer Immunology Program, Sanford Burnham Prebys Medical Discovery Institute, La Jolla, CA, USA [2]National Cancer Institute-designated Cancer Center, Sanford Burnham Prebys Medical Discovery Institute, La Jolla, CA, USA

Correspondence: zzhu@sbpdiscovery.org

models (Ersching et al, 2017). Moreover, both mice and humans with constitutive hyperactivity of PI3K show impaired humoral responses (Lucas et al, 2014; Preite et al, 2018; Wray-Dutra et al, 2018). Together, these findings suggest that tight regulation of PI3K/AKT signaling is essential for optimal GC response.

There are three AKT isoforms, AKT1, AKT2, and AKT3, which are expressed by three separate genes in a tissue-dependent manner and exhibit high homology (Hanada et al, 2004; Kumar & Madison, 2005; Manning & Toker, 2017). Notwithstanding, phenotypic analyses of AKT isoform knockout mice have shown AKT isoform–specific functions in the regulation of cellular growth, glucose homeostasis, and neuronal development (Cho et al, 2001a; Cho et al, 2001b; Easton et al, 2005). The PI3K signaling pathway plays a fundamental role in the control of B-cell growth, proliferation, and metabolism (Donahue & Fruman, 2003; Calamito et al, 2010; Jellusova & Rickert, 2016; Boothby & Rickert, 2017). GC B cells undergo rapid proliferation and require metabolic reprogramming to produce the building blocks for the synthesis of new macromolecules, as well as to meet increasing energy demands (Cho et al, 2016; Jellusova et al, 2017). Based on these facts, we hypothesize that AKT proteins may have unique and shared roles in the GC response.

Here, we provided evidence showing that B cell–intrinsic AKT1/2 are essential for GC formation and maintenance, affinity maturation, and antibody production. The defective GC phenotype in AKT1/2–deficient animals is partially attributed to impaired survival because of reduced Mcl-1 expression and diminished proliferation due to reduced c-Myc, cyclin D2, and p-S6 expression in BCR-activated AKT1/2–deficient B cells, respectively. In addition, BCR-activated AKT1/2–deficient B cells exhibited reduced glucose uptake, mitochondrial mass (MM), and oxygen consumption rate (OCR), which altogether dramatically restrict GC development. T-cell help in the form of CD40 stimulation is sufficient to restore BCR-induced proliferation and ATP production in AKT1/2–deficient B cells and rescue the loss of AKT1/2–deficient GCs in vivo. Moreover, we showed that AKT-FOXO1 axis determines activation-induced cytidine deaminase (AID)-dependent class switch recombination (CSR) and IRF4-driven plasma cell differentiation. These findings provide insights into the regulation of GC B cell fate by the PI3K/AKT pathway and how alterations in this pathway, such as the acquisition of the $Foxo1^{T24A}$ mutation, may impact B lymphomagenesis.

## Results

### B cell–intrinsic AKT1/2 function is required for GC formation, maintenance, and plasma cell differentiation

Although the three isoforms of AKT regulate cell growth, proliferation, and survival in a wide variety of cell types, their role in GC B cell response remains undefined. Because AKT deficiency affected the development of peripheral B-cell subpopulations, we crossed $Akt1^{f/f}$ (Cho et al, 2001b) and $Akt2^{-/-}$ (Dummler et al, 2006) mice or $Akt1^{f/f}$ and $Akt3^{-/-}$ (Tschopp et al, 2005) mice to mice expressing Cre recombinase in B cells activated by T cell–dependent immunization ($C\gamma1^{Cre}$) (Casola et al, 2006) to generate $C\gamma1^{Cre} × Akt1^{f/f} × Akt2^{-/-}$ or $C\gamma1^{Cre} × Akt1^{f/f} × Akt3^{-/-}$ mice. Mice were immunized with sheep red blood cells (SRBCs) on day 0 and

analyzed on day 7 to study the GC response. Remarkably, we observed ~10-fold reduction in the percentage of GC B cells in the $C\gamma1^{Cre} × Akt1^{f/f} ×$ Akt2$^{-/-}$ mice when compared with WT Ctrl ($C\gamma1^{Cre}$) or $C\gamma1^{Cre} × Akt1^{f/f} × Akt3^{-/-}$ mice (Fig 1A). The decrease in GC formation was also revealed by immunofluorescent staining of spleen sections, which showed very few GC B cells (peanut agglutinin [PNA]$^+$) in the follicles of $C\gamma1^{Cre} × Akt1^{f/f} × Akt2^{-/-}$ mice (Fig 1B). One explanation for the lack of GCs in $C\gamma1^{Cre} × Akt1^{f/f} × Akt2^{-/-}$ mice is a failure of B cells to induce or respond to Tfh cells, which are the most important T-cell subset for the GC response (Crotty, 2019). By analyzing the SRBC-immunized mice on day 7, we observed that the percentage of Tfh population in $C\gamma1^{Cre} × Akt1^{f/f} × Akt2^{-/-}$ mice was ~2.0-fold reduced when compared with the WT Ctrl ($C\gamma1^{Cre}$) mice (Fig 1C), consistent with the co-dependency of the GC B cells and Tfh cells. We also performed experiments to examine DZ and LZ GC B cells in SRBC-immunized WT Ctrl ($C\gamma1^{Cre}$) and $C\gamma1^{Cre} × Akt1^{f/f} × Akt2^{-/-}$ mice. We found that LZ GC B cells were reduced to a greater extent than DZ GC B cells in $C\gamma1^{Cre} × Akt1^{f/f} × Akt2^{-/-}$ mice, which leads to a relatively higher DZ/LZ ratio in $C\gamma1^{Cre} × Akt1^{f/f} × Akt2^{-/-}$ mice than that in WT Ctrl ($C\gamma1^{Cre}$) mice (Fig S1A and B).

The abrogation of GC B cells observed at the peak of the response in $C\gamma1^{Cre} × Akt1^{f/f} × Akt2^{-/-}$ mice could have resulted from a failure of early GC founder cells to properly mature and expand. To address this possibility, we used an analytic approach that we had previously developed involving immunization with SRBCs, which induces a robust T dependent (TD) response, with well-defined kinetics and associated surface marker expression (Cato et al, 2011). WT Ctrl ($C\gamma1^{Cre}$) and $C\gamma1^{Cre} × Akt1^{f/f} × Akt2^{-/-}$ mice were immunized with SRBCs, and GC maturation was assessed on day 7 postimmunization using PNA, GL7, and IgD. The proportion of PNA$^+$GL7$^+$IgD$^-$ mature GC B cells increased in WT Ctrl ($C\gamma1^{Cre}$) animals to comprise most GC B cells, accompanied by a concomitant decrease in PNA$^+$GL7$^-$IgD$^+$ early GC B cells. In contrast, most PNA$^+$ B cells in $C\gamma1^{Cre} × Akt1^{f/f} × Akt2^{-/-}$ mice retained the GL7$^-$IgD$^+$ phenotype, and <10% of the cells matured into GL7$^+$IgD$^-$ GC B cells (Fig 1D). Thus, these results suggest that the few PNA$^+$GL7$^-$IgD$^+$ early GC B cells that are present in $C\gamma1^{Cre} × Akt1^{f/f} × Akt2^{-/-}$ mice fail to mature and expand. Interestingly, the defect in GC maturation paralleled the reduced frequency of B220$^{lo}$CD138$^+$ plasma cells on day 7 (Fig 1E).

AKT1/2 are known regulators of B cell survival, raising the possibility that the loss of AKT1/2 may result in shortened GC B cell lifespan. To tes/t this hypothesis, we used mice expressing a tamoxifen (Tam)-inducible human CD20-driven Cre ($Tam^{Cre}$) (Khalil et al, 2012) to delete AKT1/2 in B cells after establishment of an ongoing GC reaction. Interestingly, AKT1/2 deletion resulted in depletion of GC B cells after 3 d of tamoxifen treatment (Fig 1F and G), suggesting that AKT1/2 are essential for GC B cell maintenance. Collectively, these data demonstrated that AKT1/2 are B cells intrinsically required for GC formation and maintenance.

### AKT1/2 deficiency impairs high-affinity antibody production in vivo

Because a robust GC reaction facilitates the production of high-affinity antibody, we sought to determine the role of AKT1/2 in GC B cell affinity maturation. We immunized WT Ctrl ($C\gamma1^{Cre}$) and $C\gamma1^{Cre} × Akt1^{f/f} × Akt2^{-/-}$ mice with the hapten NP (4-hydroxy-3-nitrophenylacetyl) coupled to a carrier protein CGG (chicken gamma globulin) with a

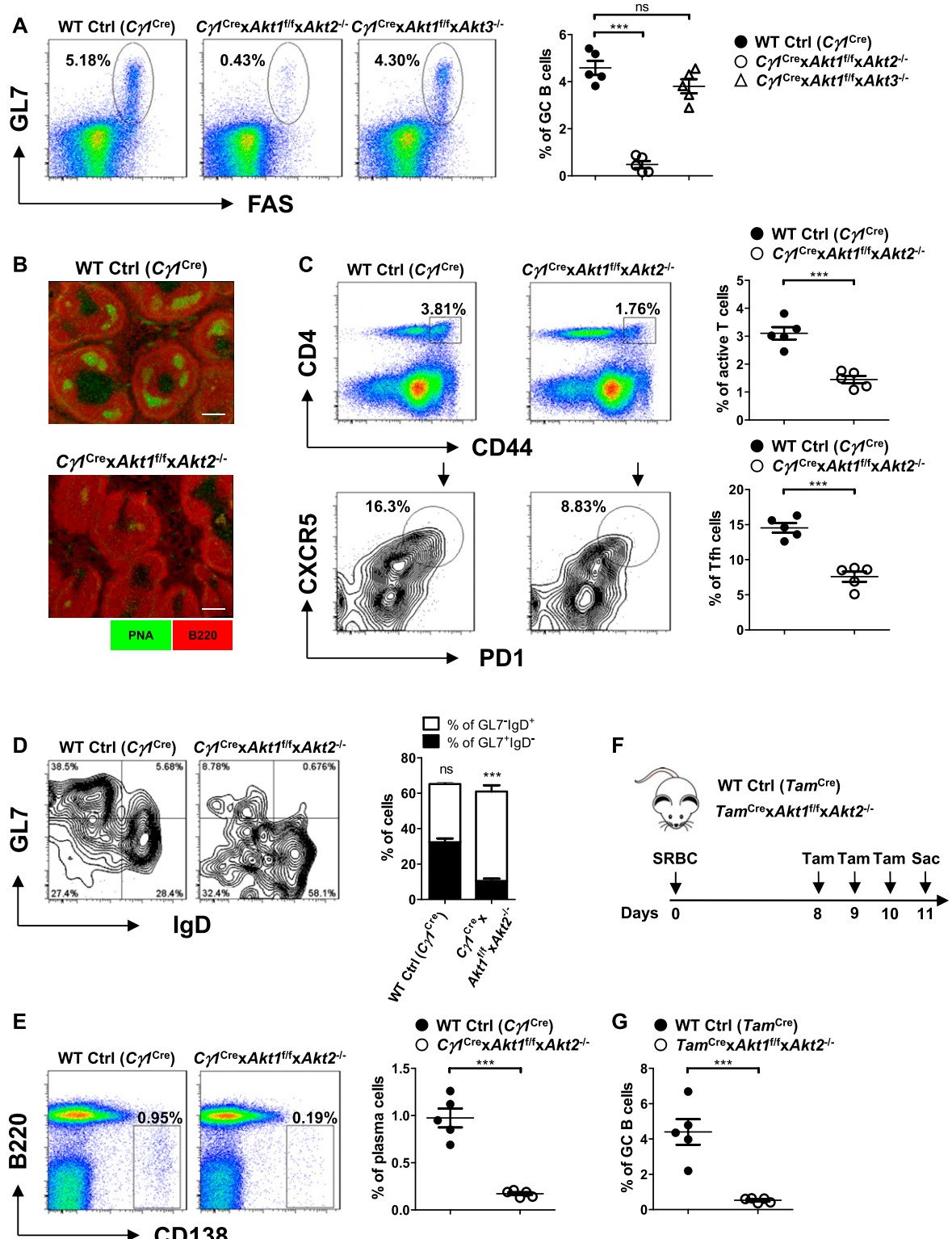

**Figure 1. AKT1/2 are required for GC response.**
**(A, B, C, D, E)** Mice of each genotype (n = 5) were immunized with SRBCs and analyzed on day 7. **(A)** Representative FACS plots (left panel) and the frequencies of GL7⁺FAS⁺ GC B cells as a percentage of total B220⁺ B cells (right panel) in the spleen are shown. **(B)** Cryosections from the spleens of each genotype of mice were immunofluorescently stained for GC B cells (PNA, in green) and follicular B cells (B220, in red). Scale bar, 100 μm. Data are representative of three independent experiments. **(C)** Representative FACS plots depicting the gating strategy for Tfh cells (left panel) and the frequencies of CD4⁺CD44⁺–active T cells (right panel, top) and CXCR5⁺PD1⁺ Tfh cells as a percentage of total CD4⁺CD44⁺–active T cells (right panel, bottom) in the spleen are shown. **(D)** Representative FACS plots showing the

conjugation ratio of 25:1 (NP25), which allows the analysis of antigen-specific immune response. Cellular responses were first analyzed on day 21 after immunization, when GC and memory B cells coexist. The frequency of antigen-specific IgG1-switched (NP⁺IgG1⁺) B cells was significantly reduced in the spleen of $Cγ1^{Cre} × Akt1^{f/f} × Akt2^{-/-}$ mice as compared with WT Ctrl ($Cγ1^{Cre}$) mice (Fig 2A and B). After subdivision of NP⁺IgG1⁺ B cells into GC (CD38⁻) and memory (CD38⁺) compartments, we found that the numbers for both NP⁺IgG1⁺ GC and memory B cells were correspondingly and significantly reduced (Fig 2C and D). Interestingly, the GC compartment was affected to a greater extent than the memory compartment when AKT1/2 were conditionally deleted, which leads to a relatively low GC/memory B cell ratio (Fig 2E). Thus, AKT1/2 are essential for the generation of antigen-specific IgG1 B cells and appears to be a limiting determinant in the magnitude of the GC response.

To investigate whether AKT1/2 deficiency affects the ability of B cells to undergo affinity maturation, we measured the titers of circulating high-affinity (NP4-binding) and total (NP23-binding) antigen-specific serum IgG1 and IgM on day 14 after NP-CGG immunization by ELISA. In contrast with WT Ctrl ($Cγ1^{Cre}$) mice, $Cγ1^{Cre} × Akt1^{f/f} × Akt2^{-/-}$ mice showed poor production of both high-affinity and total antigen-specific IgG1 antibodies and produced lower affinity antibody as indicated by the ratio of NP4-specific to NP23-specific IgG1 antibodies (Fig 2F), which serves as a measure of affinity maturation of a humoral response. However, AKT1/2 deficiency did not alter the production of high-affinity and total NP-specific IgM and their ratio (Fig 2G). To assess the role of the three isoforms of AKT in the regulation of antibody affinity maturation, we immunized WT Ctrl ($Cγ1^{Cre}$), $Cγ1^{Cre} × Akt1^{f/f}$, $Cγ1^{Cre} × Akt2^{-/-}$, and $Cγ1^{Cre} × Akt3^{-/-}$ mice with NP-CGG and measured antigen-specific serum IgG1 and IgM on day 14 postimmunization. Although AKT1, AKT2, and AKT3 single knockout mice displayed normal affinity maturation of NP-specific IgM, ablation of AKT1 or AKT2 isoform alone significantly impaired NP-specific IgG1 affinity maturation with a greater extent for AKT1 deficiency (Fig S2A and B). Together, these findings demonstrate that AKT1/2 are required for antigen-specific IgG1 affinity maturation.

### AKT1 is the predominant regulator of CSR in vitro and in vivo

Previously, we have shown that elevation of PI3K/AKT signaling through the loss of phosphatase and tensin homolog (PTEN) strongly suppresses CSR and the mechanism is directly linked to the AKT-FOXO1 axis (Anzelon et al, 2003; Omori et al, 2006; Dengler et al, 2008). To resolve the role of the three isoforms of AKT on CSR, we crossed $Akt1^{f/f}$, $Akt2^{-/-}$, and $Akt3^{-/-}$ mice to B cell–specific $CD19^{Cre}$ mice (Rickert et al, 1997). Western blot analysis confirmed purified splenic B cells from $CD19^{Cre} × Akt1^{f/f}$, $CD19^{Cre} × Akt2^{-/-}$, and $CD19^{Cre} × Akt3^{-/-}$ mice did not express AKT1, AKT2, and AKT3 protein, respectively (Fig S3A–C). In addition, Western blot analysis showed that AKT1 was expressed at higher levels than AKT2 and AKT3 (Fig S3D). We

analyzed in vitro CSR of WT Ctrl ($CD19^{Cre}$), AKT1-, AKT2-, and AKT3-deficient B cells stimulated by anti-CD40 plus IL-4, a condition that mimics signals during a TD response and favors isotype switching to IgG1. We found that the percentage of IgG1-switched B cells was increased by ~2.0-fold in AKT1-deficient B cells compared with that in WT Ctrl ($CD19^{Cre}$) B cells, whereas IgG1 switching was also enhanced in stimulated AKT2-deficient B cells, but to a lesser extent than seen in AKT1-deficient B cells (Fig 3A). Similar results were observed when B cells were activated by LPS, a condition that mimics signals during a T independent response and favors isotype switching to IgG3 (Fig 3B). We next assessed the effect of the three isoforms of AKT on CSR in vivo after immunization of WT Ctrl ($CD19^{Cre}$), $CD19^{Cre} × Akt1^{f/f}$, $CD19^{Cre} × Akt2^{-/-}$, and $CD19^{Cre} × Akt3^{-/-}$ mice with SRBCs. We found that AKT1-deficient mice displayed increased antigen-specific IgG1 titers and reciprocal decreased antigen-specific IgM titers (Fig 3C and D). In contrast with AKT1-deficient mice, AKT2- and AKT3-deficient mice showed normal production of antigen-specific IgG1 and IgM titers (Fig 3C and D).

Because AKT1/2 deficiency results in loss of GC B cells, we evaluated the effect of the three isoforms of AKT on CSR in induced GC B (iGB) cells, which can be generated using the CD40LB feeder cell line stably transfected with CD40L and B-cell activating factor (BAFF), and undergo class switching from IgM to IgG1 when induced by exogenous IL-4 (Nojima et al, 2011). When naive WT Ctrl ($CD19^{Cre}$) B cells were cultured on the 40LB feeder cells, they underwent massive expansion and acquired the GL7⁺FAS⁺ GC phenotype. We were able to efficiently induce AKT1-, AKT2-, AKT3-, AKT1/2-, and AKT1/3-deficient B cells to differentiate into GL7⁺FAS⁺ GC-like B cells after 5 d in culture with 40LB cells (Fig 3E, top) which enabled us to carefully compare the role of different isoforms of AKT on CSR. The percentage of IgG1-switched iGB cells derived from AKT1-deficient B cells and AKT2-deficient B cells was markedly increased by ~2.5-fold and ~2.0-fold, respectively, compared with that from WT Ctrl ($CD19^{Cre}$) B cells (Fig 3E, bottom). The percentage of IgG1-switched iGB cells derived from AKT1/2-deficient B cells was ~1.3-fold and ~1.7-fold higher than that from AKT1- and AKT2-deficient B cells alone, respectively (Fig 3E, bottom), suggesting that AKT1 and AKT2 both repress CSR. The percentage of IgG1-switched iGB cells derived from AKT3-deficient B cells was slightly higher than that from WT Ctrl ($CD19^{Cre}$) B cells (Fig 3E, bottom), suggesting that AKT3 appears to have little effect on CSR, which was consistent with the comparable percentage of IgG1-switched iGB cells-derived from AKT1/3-deficient and AKT1-deficient B cells (Fig 3E, bottom). Together, these data demonstrate that AKT1 is the most important isoform for in vitro and in vivo CSR.

### Foxo1^{T24A} increases in vitro and in vivo CSR in B cells

PI3K-mediated AKT activation leads to phosphorylation of FOXO1 (at residues threonine 24 [T24], and serine 256 [S256] and 319 [S319]),

progression of GCs in the splenic B220⁺PNA⁺ populations (the number of events for WT Control ($Cγ1^{Cre}$) versus $Cγ1^{Cre} × Akt1^{f/f} × Akt2^{-/-}$: 825 versus 251) (left panel). The frequencies of early GL7⁻IgD⁺ and mature GL7⁺IgD⁻ GC B cells were plotted (right panel). **(E)** Representative FACS plots for B220⁺CD138⁺ plasma cells in the splenic lymphocytes gate (left panel). The frequencies of plasma cells were plotted (right panel). **(F)** Time line illustration of tamoxifen treatment for SRBC-immunized WT Ctrl ($Tam^{Cre}$) and $Tam^{Cre} × Akt1^{f/f} × Akt2^{-/-}$ mice. The mice were immunized with SRBCs on day 0, orally gavaged with tamoxifen on day 8, 9, and 10, and analyzed on day 11 postimmunization. **(G)** GC B cells in the spleen of mice in (F) were gated as B220⁺GL7⁺FAS⁺ cells. The frequencies of GC B cells are shown. Two-tailed $t$ tests were used to test statistical significance for (C–E and G). One-way ANOVA was used to test statistical significance for (A). Symbols represent individual mice studied. Error bars represent mean ± SEM. ***$P$ < 0.001.

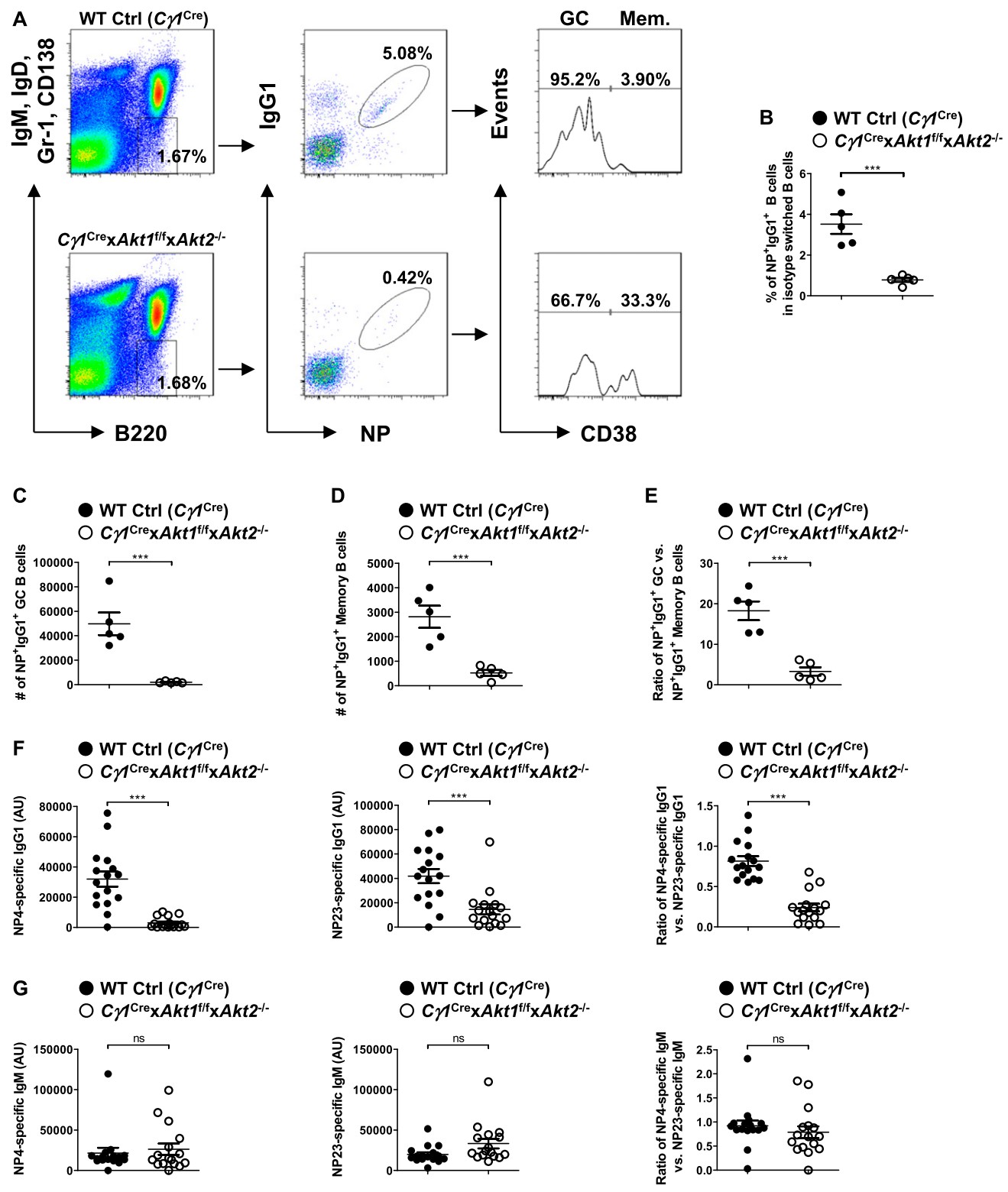

**Figure 2. Conditional deletion of AKT1/2 abrogates formation of NP-specific GC B cells and compromises antibody affinity maturation.**
**(A)** Representative FACS plots for analysis of splenocytes on day 21 after i.p.-immunization with NP-CGG in alum. Isotype-switched B cells (IgM⁻IgD⁻Gr-1⁻CD138⁻B220⁺) were analyzed for NP⁺IgG1⁺ status. NP⁺IgG1⁺ B cells were further subdivided into GC (CD38⁻) or memory (CD38⁺) B cells. **(B)** Frequencies of the NP⁺IgG1⁺ B cell populations identified in (A) are shown for five mice in each group. **(A, C, D, E)** Absolute cell numbers of the NP⁺IgG1⁺CD38⁻ GC B populations (C) and the NP⁺IgG1⁺CD38⁺ memory B populations (D) identified in (A), and their ratios (E) are shown for five mice in each group. **(F, G)** Production of high-affinity (NP4 binding) and total (NP23 binding) NP-specific IgG1 (F) and IgM (G) was measured by ELISA on day 14 after NP-CGG immunization. The ratios of NP4- versus NP23-specific IgG1 (F) and IgM (G) were also shown.

mediating its interaction with 14-3-3 protein and the nuclear export of the complex (Tzivion et al, 2011). We have previously shown that the AKT-FOXO1 axis controls CSR via induction of AID expression in mature B cells (Omori et al, 2006; Dengler et al, 2008). However, in GC B cells, FOXO1 regulates CSR via facilitating AID targeting to particular switch regions without affecting AID protein expression (Dominguez-Sola et al, 2015; Sander et al, 2015). The degree to which FOXO1 inactivation accounts for AKT function in B cells is unclear. To approach this question, we generated a novel mouse model, which inducibly expresses the $Foxo1^{T24A}$ mutant (Fig S4). We first determined FOXO1 subcellular localization in $Foxo1^{WT/WT}$ and $Foxo1^{T24A/T24A}$ B cells in the resting state. We found that, in both $Foxo1^{WT/WT}$ and $Foxo1^{T24A/T24A}$ B cells, FOXO1 was mainly located in the nucleus with less in the cytoplasm (Fig 4A). Phosphorylation of FOXO1 at a known AKT-phosphorylation site (S256) was strong in both $Foxo1^{WT/WT}$ and $Foxo1^{T24A/T24A}$ B cells. p-FOXO1 (S256) was largely restricted to the cytoplasm of $Foxo1^{WT/WT}$ B cells, whereas it was detected primarily in the nucleus of $Foxo1^{T24A/T24A}$ B cells (Fig 4A). Next, we examined the p-FOXO1 (S256) subcellular localization upon anti-BCR or anti-CD40 stimulation. We observed that the BCR or CD40 induced strong p-AKT (S473) in $Foxo1^{WT/WT}$ B cells 10 min after stimulation and its subsequent translocation from cytoplasm to nucleus. Higher levels of nuclear AKT activity correlated with increased activation of p-FOXO1 (S256) in the nucleus (10 min after stimulation) and p-FOXO1 (S256) nuclear export followed by subsequent degradation in the cytoplasm (30 min after stimulation) (Fig 4A). In contrast to the findings in $Foxo1^{WT/WT}$ B cells, in $Foxo1^{T24A/T24A}$ B cells p-FOXO1 (S256) was restricted to the nucleus (Fig 4A).

To determine the functional consequences of altered FOXO1 distribution, we analyzed in vitro CSR efficiency of $Foxo1^{WT/WT}$, $Foxo1^{WT/T24A}$, $Foxo1^{T24A/T24A}$, and FOXO1-deficient B cells upon anti-CD40 and IL-4 stimulation. We found that ~15% of $Foxo1^{WT/T24A}$ and ~59% of $Foxo1^{T24A/T24A}$ B cells were positive for IgG1, respectively, in striking contrast to ~6.0% of IgG1$^+$ cells in $Foxo1^{WT/WT}$ B cells and a nearly complete block in IgG1 switching in FOXO1-deficient B cells (Fig 4B). These results suggest that CSR is controlled by nuclear FOXO1 in a dose-dependent manner. We further analyzed in vivo CSR in SRBC-immunized $Cγ1^{Cre} × Foxo1^{WT/WT}$, $Cγ1^{Cre} × Foxo1^{WT/T24A}$, and $Cγ1^{Cre} × Foxo1^{T24A/T24A}$ mice. We found that the proportion of the GC B cells stained positive for IgG1 was larger in both $Cγ1^{Cre} × Foxo1^{WT/T24A}$ and $Cγ1^{Cre} × Foxo1^{T24A/T24A}$ groups than that in $Cγ1^{Cre} × Foxo1^{WT/WT}$ group (Fig 4C). We previously reported that mature B cells lacking FOXO1 had severe defects in class switching due to failed up-regulation of AID expression (Dengler et al, 2008). Here, we tested if $Foxo1^{T24A/T24A}$ increased in vitro CSR via induction of AID expression. We investigated the corresponding cultures in Fig 4B via Western blot for AID, p53, and γH2A.X. We found that AID expression was significantly elevated in $Foxo1^{T24A/T24A}$ B cells compared with that in $Foxo1^{WT/WT}$, $Foxo1^{WT/T24A}$, and FOXO1-deficient B cells (Fig 4D). Enhanced AID expression creates DNA damage in B cells, which results in activation of p53 pathway (Daniel & Nussenzweig, 2013; Casellas et al, 2016). In agreement with this notion, $Foxo1^{T24A/T24A}$ B cells expressed highly increased p53 and γH2A.X as a marker for DNA damage (Fig 4D). In

contrast, we did not detect γH2A.X expression in FOXO1-deficient B cells (Fig 4D). Together, these results indicate that nuclear FOXO1 promotes CSR via up-regulation of AID expression as the primary mechanism.

## AKT-FOXO1 axis instructs plasma cell differentiation in B cells

The abrogation of plasma cell differentiation in $Cγ1^{Cre} × Akt1^{f/f} × Akt2^{-/-}$ mice (Fig 1E) prompted us to assess the role of the three isoforms of AKT in B-cell differentiation. We analyzed plasma cell differentiation of WT Ctrl ($CD19^{Cre}$), AKT1-, AKT2-, and AKT3-deficient B cells cultured in vitro with LPS using flow cytometry. We observed ~40% reduction in plasma cell differentiation in LPS-stimulated AKT1-deficient B cells compared with that in LPS-stimulated WT Ctrl ($CD19^{Cre}$) B cells, whereas plasma cell differentiation was slightly diminished in LPS-stimulated AKT2- or AKT3-deficient B cells (Fig 5A). Intriguingly, these changes in differentiation did not correlate with cell proliferation because AKT1-, AKT2-, and AKT3-deficient B cells exhibited no difference in the number of cell divisions induced by LPS stimulation (Fig 5B).

To understand how AKT1 regulates plasma cell differentiation, we analyzed plasma cell differentiation of B cells isolated from WT Ctrl ($Tam^{Cre}$), $Tam^{Cre} × Akt1^{f/f}$, $Tam^{Cre} × Akt1^{f/f} × Foxo1^{f/f}$ mice after tamoxifen treatment. AKT1-deficient B cells developed into fewer plasma cells than WT Ctrl ($Tam^{Cre}$) B cells upon stimulation with LPS, which can be fully rescued by concomitant loss of FOXO1 (Fig 5C). This result supports our previous report examining the AKT-FOXO1 axis in plasma cell differentiation (Omori et al, 2006). The initiation of plasma cell differentiation requires the dual-regulation of transcription factors, IRF4 and PAX5 (Nutt et al, 2015). Therefore, we analyzed IRF4 and PAX5 expression in LPS-stimulated B cells from WT Ctrl ($Tam^{Cre}$) and $Tam^{Cre} × Foxo1^{f/f}$ mice by flow cytometry. A small subset of WT Ctrl ($Tam^{Cre}$) B cells adopted the typical plasma cell (IRF4$^{hi}$PAX5$^{lo}$) signature, whereas in FOXO1-deficient B cells, this population was increased by ~2.0-fold (Fig 5D). Using lysates from corresponding cell cultures, we immunoblotted for the expression of the transcription factors IRF4, BLIMP1, PAX5, and BACH2 and found that FOXO1-deficient B cells showed dramatically increased the up-regulation of IRF4 compared with WT Ctrl ($Tam^{Cre}$) B cells (Fig 5E). Conversely, BLIMP1 and PAX5 were normally up-regulated and down-regulated, respectively, in FOXO1-deficient B cells (Fig 5E). In addition to PAX5, BACH2 is another transcriptional repressor of BLIMP1 and shutting down expression of BACH2 appears to be crucial to establish the plasma cell program (Muto et al, 2010). However, we found that BACH2 expression was unaltered in the presence or absence of FOXO1 (Fig 5E). Thus, these findings suggest that FOXO1 deficiency promotes plasma cell differentiation through induction of IRF4 expression. Consistent with these findings, LPS-stimulated FOXO1-deficient B cells showed increased IgM production relative to WT Ctrl ($Tam^{Cre}$), as measured by ELISA from cell culture supernatants (Fig 5F). Moreover, the baseline levels of serum IgM in $Tam^{Cre} × Foxo1^{f/f}$ mice were higher than that in WT Ctrl ($Tam^{Cre}$) mice (Fig 5G). Together, these findings demonstrate that

---

Plots were representative of 16 WT Ctrl ($Cγ1^{Cre}$) and 16 $Cγ1^{Cre} × Akt1^{f/f} × Akt2^{-/-}$ mice. Two-tailed $t$ tests were used to test statistical significance for (B, C, D, E, F, G). Symbols represent individual mice studied. Error bars represent mean ± SEM. ***$P < 0.001$.

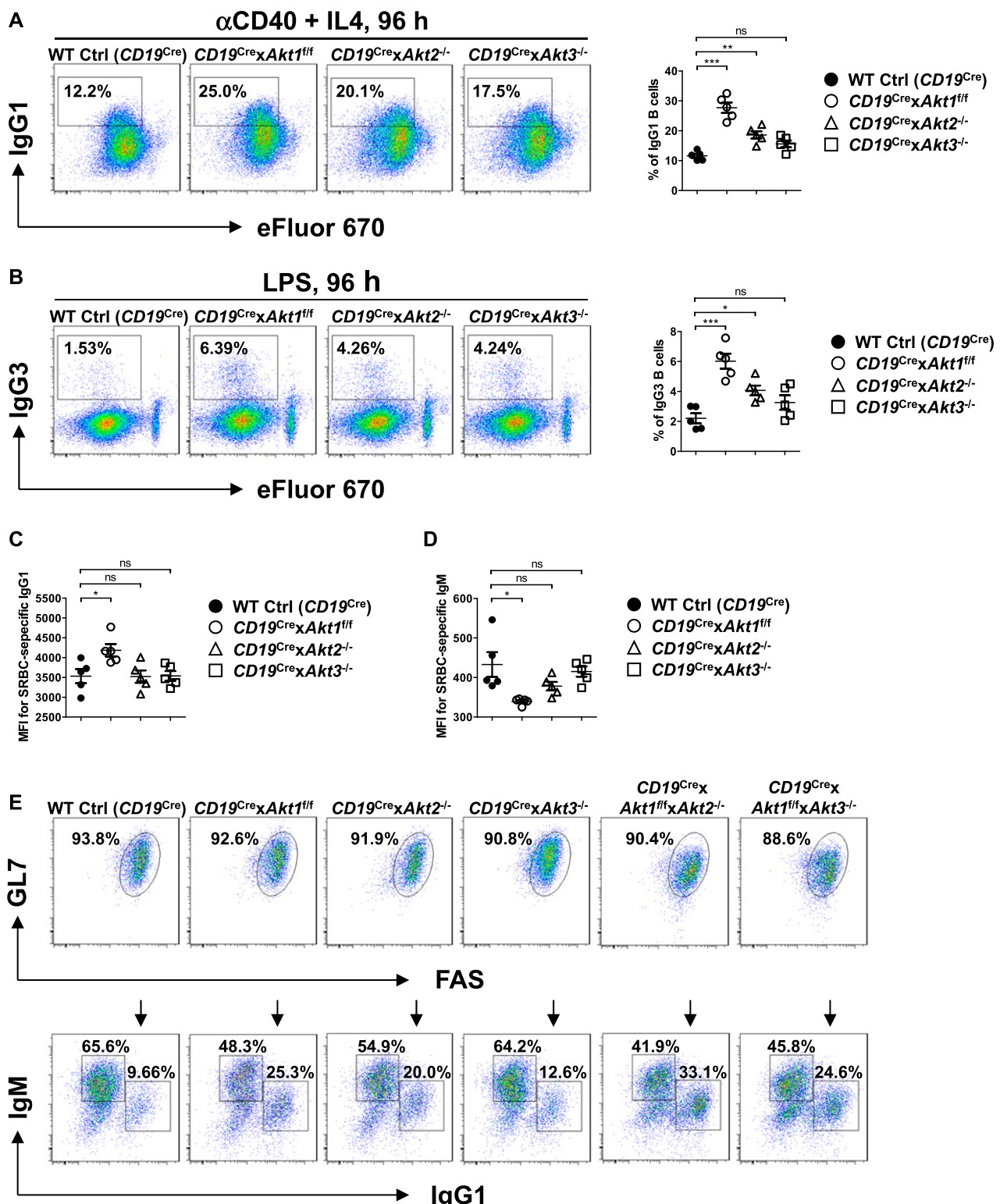

**Figure 3. AKT1 predominantly regulates in vitro and in vivo CSR.**
**(A, B)** Representative FACS plots showing IgG1 class switch (A) or IgG3 class switch (B) in eFluor 670-labeled splenic B cells purified from mice of each genotype (n = 5) stimulated with anti-CD40 plus IL4 or LPS for 4 d, respectively (left panel). The frequencies of IgG1⁺ or IgG3⁺ B cells were plotted (right panel). **(C, D)** Sera were collected from SRBC-immunized mice (n = 5) on day 7 after immunization. Serum levels of SRBC-specific IgG1 (C) and IgM (D) antibody were measured by flow cytometry staining. Plots show the obtained mean fluorescence intensities (MFIs). **(E)** Purified splenic B cells from mice of each genotype were cultured with IL-4 on 40LB feeder cells for 5 d to allow for induction of phenotypically GC B cells. The iGB cells were characterized by flow cytometry analysis of surface markers GL7 and FAS expression (top). IgM and

plasma cell differentiation is determined by the degree of PI3K/AKT signaling, and the level of nuclear FOXO1 activity.

## AKT1/2 are required for BCR-activated B-cell survival, growth, and proliferation

We then aimed to understand how AKT1/2 loss impaired the GC B-cell response. We first analyzed whether or not AKT1/2 deficiency affected B-cell proliferation, which is a critical event for GC B-cell response. A previous study showed that AKT1/2-deficient follicular B cells proliferated robustly, comparable with WT Ctrl B cells, in response to LPS but were hyporesponsive to anti-IgM stimulation (Calamito et al, 2010). To investigate the nature of this proliferative defect, we performed cell cycle analysis using BrdU incorporation in combination with the DNA dye 7-AAD and observed a significant reduction in the S-phase entry of AKT1/2-deficient B cells 24 h after BCR engagement (Fig 6A). There was also a reduction in the S-phase entry of AKT1/3-deficient B cells, but to a lesser extent than that of AKT1/2-deficient B cells (Fig 6A). To rule out the possibility that AKT1/2-deficient B cells are simply unresponsive to BCR cross-linking, the levels of various activation markers on the cell surface were assessed by flow cytometry. When stimulated with anti-IgM, AKT1/2-deficient B cells up-regulated CD69, CD80, CD86, and MHC II to levels comparable with WT Ctrl (*CD19*^Cre) B cells (Fig S5). BCR stimulation induces expression of both cyclin D2 and cyclin D3 in normal B cells (Solvason et al, 1996). The two D-type cyclins are capable of forming active p-Rb kinase complexes, thus freeing the E2F protein to drive transcription of genes required for the transition from the G1 to S phase. In line with the cell cycle analysis, we observed less induction of cyclin D2, but not cyclin D3, in both AKT1/2-deficient and AKT1/3-deficient B cells than that in WT Ctrl (*CD19*^Cre) B cells 24 h after anti-IgM treatment (Fig 6B). The *Myc* gene, which encodes a transcription factor that regulates G1 to S phase progression, is expressed at a high level in a subset of GC B cells (Calado et al, 2012; Dominguez-Sola et al, 2012). Interestingly, baseline levels of c-Myc expression is much lower in AKT1/2-deficient B cells than that in WT Ctrl (*CD19*^Cre) and AKT1/3-deficient B cells (Fig 6B). Furthermore, c-Myc induction upon anti-IgM stimulation was insufficient in AKT1/2-deficient B cells (Fig 6B), consistent with the role of AKT in the inactivation of GSK3-mediated degradation of c-Myc (Diehl et al, 1998; Gregory et al, 2003). Survival of GC B cells is mediated by both the intrinsic and extrinsic apoptotic cell death pathways (Strasser et al, 2009). Given that the anti-apoptotic molecules Mcl-1 and Bcl-xL are critical for GC B cell survival (Vikstrom et al, 2010), we determined the impact of AKT loss on their expression. We found that Mcl-1 induction, but not Bcl-xL, was significantly reduced in BCR-stimulated AKT1/2-deficient B cells compared with that in BCR-stimulated WT Ctrl (*CD19*^Cre) and AKT1/3-deficient B cells (Fig 6B), which is consistent with previous findings showing that Mcl-1 and not Bcl-xL was indispensable for the formation and persistence of GCs (Vikstrom et al, 2010). Together, these results suggest that AKT1/2 promote BCR-induced proliferation potentially via up-regulation of cyclin D2 and c-Myc and are required for the survival of BCR-activated B cells via selective Mcl-1 induction.

In addition to c-Myc, p-S6 is also critical for regulating cell growth and protein synthesis. Ersching et al revealed that phosphorylation of S6 and mTOR1 activation were associated with GC B cell growth and positive selection in response to T-cell help signals (Ersching et al, 2017). To assess the role of AKT1/2 in the regulation of B cell growth, cell size was compared between B cells isolated from WT Ctrl (*CD19*^Cre) and *CD19*^Cre × *Akt1*^f/f × *Akt2*^−/− mice before and after BCR stimulation. In the resting state, AKT1/2-deficient B cells were comparable in size with WT Ctrl (*CD19*^Cre) B cells (Fig 6C). After anti-IgM stimulation, the size of both AKT1/2-deficient and WT Ctrl (*CD19*^Cre) B cells increased, whereas BCR-stimulated AKT1/2-deficient B cells were significantly smaller than WT Ctrl (*CD19*^Cre) B cells (Fig 6C), indicating that AKT1/2 regulate cell growth upon BCR stimulation. We then measured phosphorylation of the ribosomal S6 protein at the Ser-240/244 site, which is a sensitive readout of mTORC1 activity. Consistently, a significantly reduced but not complete reduction in phosphorylation of S6 was present in BCR-stimulated AKT1/2-deficient B cells based on intracellular flow and Western blot analysis (Fig 6D and E). Phosphorylation of S6 was also reduced in AKT1/3-deficient B cells, albeit to a lesser extent than that in AKT1/2-deficient B cells (Fig 6D and E). The reduced phosphorylation of S6 upon BCR stimulation in AKT1/2-deficient B cells could be due to aberrant proximal BCR signaling in these cells. To test this possibility, B cells isolated from WT Ctrl (*CD19*^Cre), *CD19*^Cre × *Akt1*^f/f × *Akt2*^−/−, and *CD19*^Cre × *Akt1*^f/f × *Akt3*^−/− mice were stimulated with anti-IgM. We found that both BLNK (Y84) and p-PLCγ2 (Y759) were strongly phosphorylated in WT Ctrl (*CD19*^Cre), AKT1/2-, and AKT1/3-deficient B cells (Fig S6A and B). Phosphorylation of Erk1/2 (T202/Y204) was robust in AKT1/2-deficient B cells, whereas it was clearly attenuated in AKT1/3-deficient B cells (Fig S6C and D), suggesting that AKT3, not AKT1 and AKT2, specifically regulates BCR-mediated Erk1/2 activation via an unknown mechanism that was further confirmed by Western blot analysis using WT Ctrl (*CD19*^Cre), AKT1-, AKT2, and AKT3-deficient B cells (Fig S6E). Collectively, these results indicate that AKT1/2 are essential for BCR-mediated B-cell growth via their effector p-S6.

## *Bcl2* transgene expression cannot rescue the loss of AKT1/2-deficient GC B cells in vivo

Given that Akt1/2-deficient mature B cells exhibited a profound survival defect, we sought to determine whether ectopic expression of Bcl2 could rescue the impaired GC response in AKT1/2-deficient mice. To investigate this matter, we crossed *CD19*^Cre × *Akt1*^f/f × *Akt2*^−/− mice with *Bcl2* transgenic mice which constitutively express the human *Bcl2* transgene in the B lineage (Strasser et al, 1991). We observed that ectopic expression of Bcl2 in *Bcl2* Tg × *CD19*^Cre × *Akt1*^f/f × *Akt2*^−/− mice was able to significantly increase the percentage of follicular B cells in the spleen (Fig 7A, top) and mature recirculating B cells in the BM (Fig S7A and B). Whereas the percentage of GC B cells was generally higher in the *Bcl2* transgenic mice because of enhanced

IgG1 on iGB cells are shown (bottom). Data are representative of three independent experiments. One-way ANOVA was used to test statistical significance for (A, B, C, D). Symbols represent individual mice studied. Error bars represent mean ± SEM. *P < 0.05; **P < 0.01; ***P < 0.001.

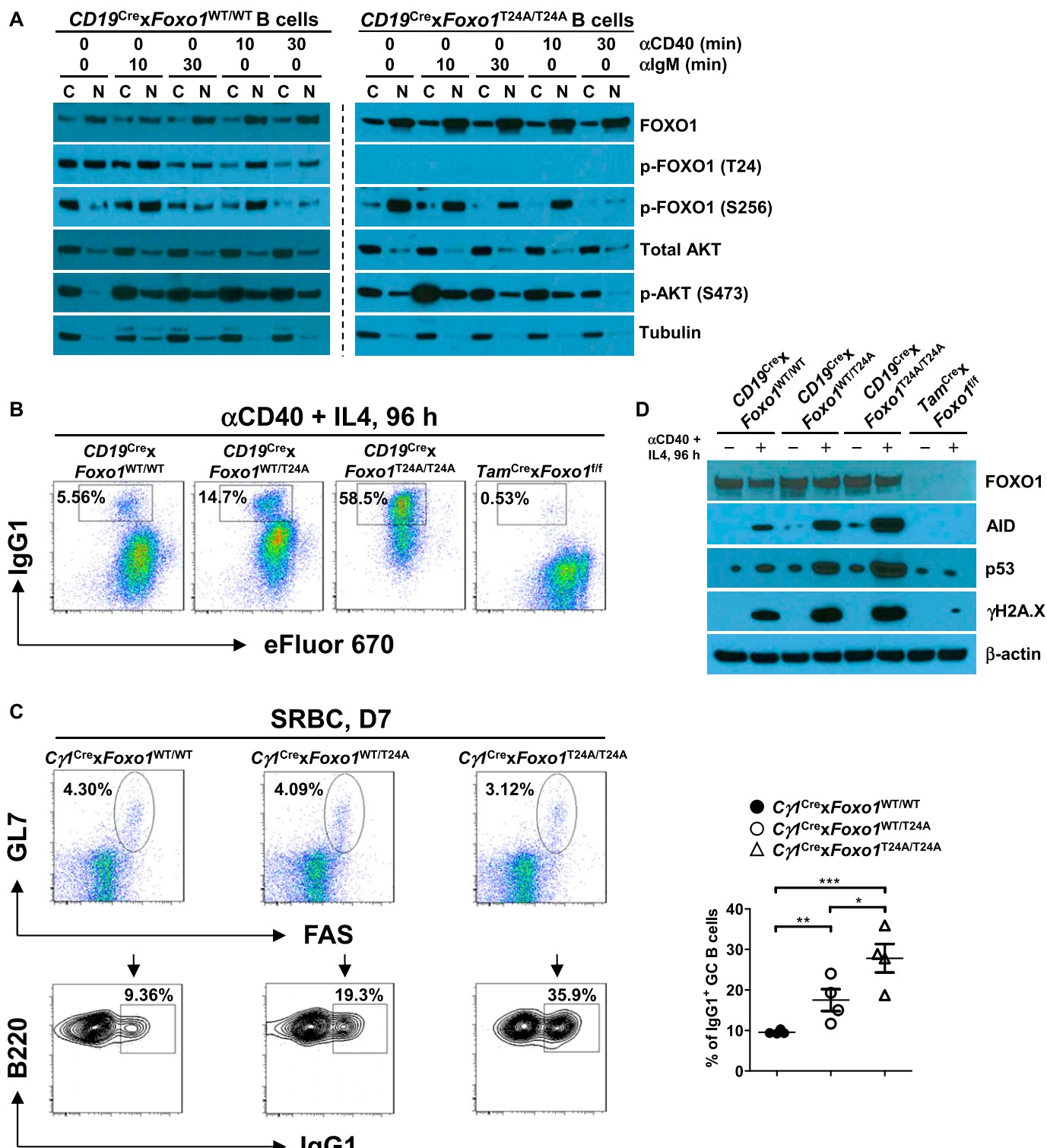

**Figure 4. Nuclear FOXO1 promotes CSR via up-regulation of AID expression.**

**(A)** Western blot analysis of FOXO1 and AKT expression in subcellular fractions of purified Foxo1[WT/WT] and Foxo1[T24A/T24A] splenic B cells unstimulated or stimulated with anti-IgM or anti-CD40 for indicated time, respectively. Tubulin was used as loading control for the cytoplasmic fraction. Data are representative of two independent experiments. **(B)** Representative FACS plots showing IgG1 class switch in eFluor 670–labeled splenic B cells purified from mice of each genotype stimulated with anti-CD40 plus IL4 for 4 d. Data are representative of three independent experiments. **(C)** Cγ1[Cre] × Foxo1[WT/WT], Cγ1[Cre] × Foxo1[WT/T24A], and Cγ1[Cre] × Foxo1 [T24A/T24A] mice (n = 4 for each genotype) were immunized with SRBCs and analyzed on day 7 postimmunization. Representative FACS plots depicting the gating strategy for IgG1 class-switched GC B cells (IgG1+B220+GL7+FAS+) in the spleen (left panel). The frequencies of IgG1 class-switched GC B cells in the spleen were assessed (right panel). **(D)** Western blot analysis of

B-cell survival, the fraction of GC B cells in *Bcl2* Tg × *CD19*[Cre] × *Akt1*[f/f] × *Akt2*[−/−] mice remained significantly lower than that in WT Ctrl (*CD19*[Cre]) mice on day 7 after SRBC immunization (Fig 7A, bottom). These findings indicate that the observed loss of AKT1/2-deficient GC B cells is not due to the inability of these cells to transmit pro-survival signals. The effect on plasma cell differentiation was also assessed where it was found that Bcl2 overexpression was able to partially rescue the loss of plasma cells in AKT1/2-deficient animals probably because of its critical role in countering high levels of Bim expression in plasma cells (Fig 7B).

### T-cell help can rescue the defective GC response in AKT1/2-deficient mice

During TD immune responses, antigen-activated B cells migrate to the B-T zone boundary to seek help from cognate T helper cells. After obtaining T-cell help, largely via CD40/CD40L interactions, B cells undergo sustained and rapid proliferation which is critical for GC formation. BCR-activated AKT1/2-deficient B cells are less proliferative (Fig 6A) despite intact proximal BCR signaling (Fig S6). AKT1/2-deficient mice showed a severe reduction in Tfh cells (Fig 1C) and impaired GC B cell expansion (Fig 1D). Nonetheless, profiling of activation markers on BCR-activated AKT1/2-deficient B cells suggests that they retain the ability to prime T cells (Fig S5). These findings prompted us to examine the fate of AKT1/2-deficient GC B cells upon receiving T-cell help. To test this possibility, we i.v.-injected *Cγ1*[Cre] × *Akt1*[f/f] × *Akt2*[−/−] mice with agonistic anti-CD40 antibody or corresponding isotype control and i.p.-immunized the mice with SRBCs on day 0 and analyzed the GC response on day 7. We found that in vivo anti-CD40 administration increased B cell numbers and significantly rescued the loss GC B, Tfh, and plasma cells in the spleen of *Cγ1*[Cre] × *Akt1*[f/f] × *Akt2*[−/−] mice (Fig 8A). Based on these results, we postulated that signaling via CD40 could rescue impaired BCR-induced proliferation in AKT1/2-deficient B cells. In keeping with this notion, we found that anti-CD40 stimulation was able to partially restore reduced BCR-induced proliferation in AKT1/2-deficient B cells (Fig 8B). To gain insight into the mechanisms of how signaling via CD40 corrected the proliferation defect in AKT1/2-deficient B cells, we examined the induction of c-Myc, cyclin D2, and p-S6 (S240/244) in AKT1/2-deficient B cells upon anti-CD40, anti-IgM, and combined stimulation. Consistent with our findings in Fig 6B and E, anti-IgM stimulation alone poorly induced c-Myc, cyclin D2, and p-S6 (S240/244) expression in AKT1/2-deficient B cells. However, combined anti-CD40 and anti-IgM stimulation strongly and synergistically induced c-Myc and p-S6 (S240/244), whereas dual signals failed to induce cyclin D2 (Figs 8C and S8). We further confirmed robust c-Myc induction in AKT1/2-deficient B cells in response to combined anti-CD40 and anti-IgM stimulation by flow cytometry (Fig 8D). As anti-CD40 and anti-BCR stimulation preferentially induces cyclin D3 versus cyclin D2 expression (Lam et al, 2000), the former mechanism likely accounts for the rescue. We found that anti-CD40 stimulation alone was able to induce cyclin D3 expression in AKT1/2-deficient B cells and combined anti-CD40 and anti-IgM stimulation had

synergistic effect on cyclin D3 expression (Fig S9). Together, these findings suggested that T-cell help can rescue the loss of AKT1/2-deficient GC B cells via induction of p-S6, c-Myc, and cyclin D3.

c-Myc and the PI3K/AKT play very important roles in the metabolic reprogramming of B cells following antigen engagement and in preparation for the energetic and metabolic requirements of proliferation and differentiation. The metabolic changes in BCR-activated B cells include increased glucose uptake, mitochondrial biosynthesis, glycolysis, and oxidative phosphorylation. We first examined the glucose uptake in WT Ctrl (*CD19*[Cre]) and AKT1/2-deficient B cells in the resting state or stimulated with anti-IgM, anti-IgM plus anti-CD40, or LPS. We found that AKT1/2-deficient B cells exhibited decreased glucose uptake, but normal expression of the glucose transporter GLUT1, comparable with WT Ctrl (*CD19*[Cre]) B cells, in response to anti-IgM or combined anti-IgM and anti-CD40 stimulation (Fig S10A and B), indicating that AKT1/2 regulate BCR-driven glucose uptake via other isoforms of GLUT. Interestingly, both glucose uptake and expression of GLUT1 were comparable between WT Ctrl (*CD19*[Cre]) and AKT1/2-deficient B cells in response to LPS stimulation (Fig S10A and B). To examine mitochondrial function, we evaluated total MM. Consistent with the known role of c-Myc in supporting mitochondrial biogenesis, the MM in AKT1/2-deficient B cells was lower than that in WT Ctrl (*CD19*[Cre]) cells after anti-IgM stimulation whereas combined anti-IgM and anti-CD40 stimulation significantly restored the MM in AKT1/2-deficient B cells (Fig 8E). The MM was comparable between LPS-stimulated WT Ctrl (*CD19*[Cre]) and AKT1/2-deficient B cells (Fig 8E). To determine whether or not AKT1/2-deficiency affected mitochondrial respiration, we measured OCR, an indicator of oxidative phosphorylation, upon addition of inhibitors of the respiratory chain with an extracellular flux analyzer. The addition of oligomycin resulted in a drop in the OCR and the subsequent addition of the uncoupler carbonyl cyanide p-trifluoro-methoxyphenyl hydrazone (FCCP) depolarized the mitochondrial which resulted in an increase in the OCR. The difference between the OCR at baseline and the OCR after the addition of FCCP represents the reserve oxidative phosphorylation capacity of the cells. We found, unlike WT Ctrl (*CD19*[Cre]) B cells, that AKT1/2-deficient B cells did not increase oxygen consumption (Fig 8F, basal respiration) to cover ATP demands in BCR-activated B cells, and their reserve capacity was diminished in BCR-activated conditions (Fig 8F, reserve capacity). These results showed that mitochondrial function is impaired in B cells lacking AKT1/2 and indicated that these cells are likely incapable of meeting the energy demands incurred during BCR activation. In line with the MM analysis, we observed combined anti-IgM and anti-CD40 stimulation dramatically increased both basal respiration and reserve capacity in AKT1/2-deficient B cells (Fig 8F) probably via their synergistic effect on c-Myc expression.

## Discussion

Previous studies showed that AKT1/2 are essential for the survival and homeostatic proliferation of mature B cells. Here, we took

---

FOXO1, AID, p53, and γH2A.X expression in purified splenic B cells from mice of each genotype unstimulated or stimulated with anti-CD40 plus IL4 for 4 d. Data are representative of three independent experiments. One-way ANOVA was used to test statistical significance for (C). Symbols represent individual mice studied. Error bars represent ± SEM. *$P$ < 0.05; **$P$ < 0.01; ***$P$ < 0.001.

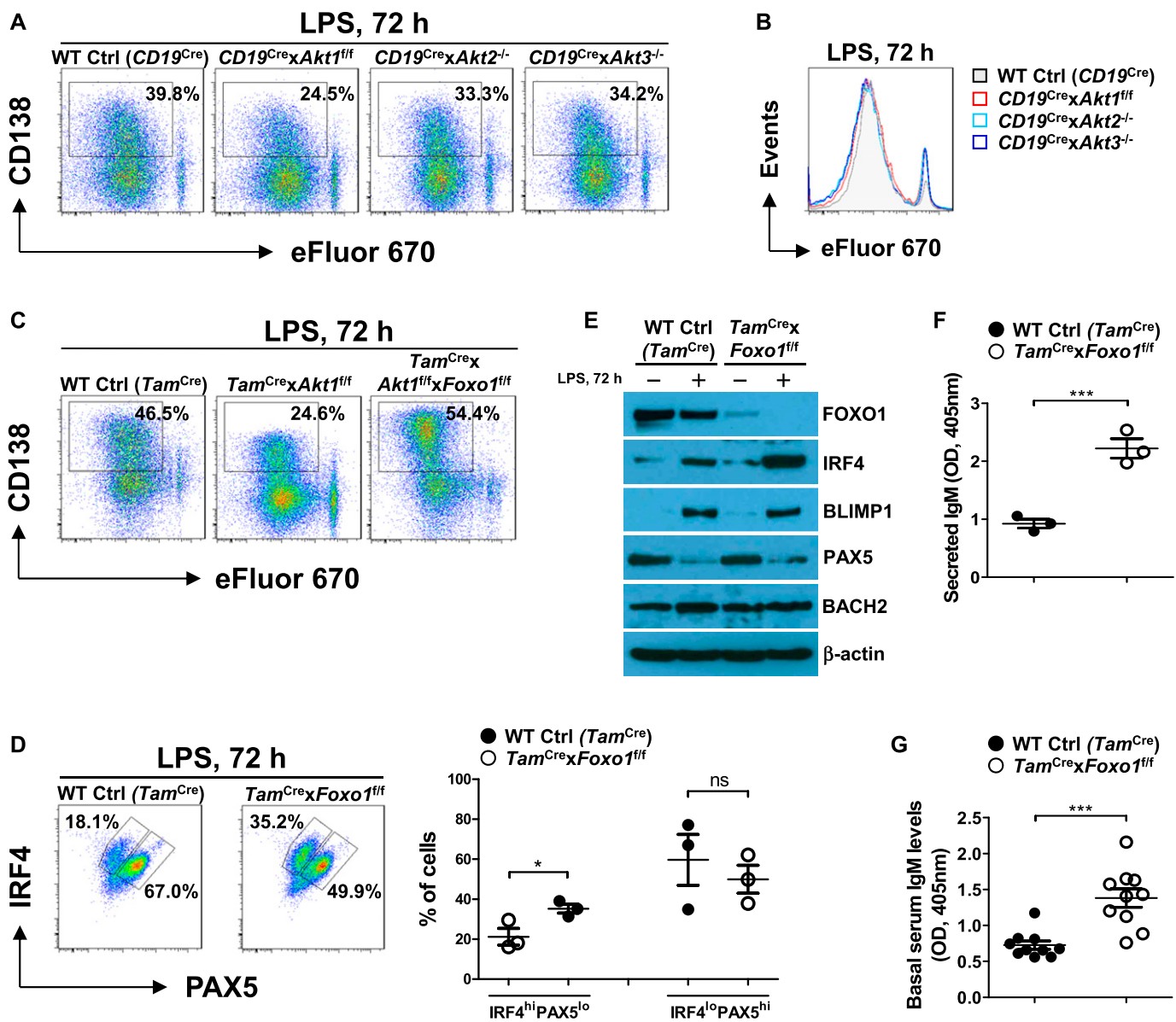

**Figure 5. AKT-FOXO1 axis promotes plasma cell differentiation via up-regulation of IRF4 expression.**
**(A)** Representative FACS plots show CD138⁺ plasma cell differentiation in purified splenic B cells from mice of each genotype stimulated with LPS for 3 d. B cells were labeled with eFluor 670 to track division history. Data are representative of three independent experiments. **(B)** Representative histogram shows eFluor 670 dilutions of B cells in (A). Data are representative of three independent experiments. **(C)** Representative FACS plots of CD138⁺ plasma cells and the cell division history of eFluor 670-labeled splenic B cells from tamoxifen-treated mice of each genotype stimulated with LPS for 3 d. Data are representative of three independent experiments. **(D)** Representative FACS plots for expressions of IRF4 and PAX5 as determined using flow cytometry of purified splenic B cells from tamoxifen-treated mice of each genotype stimulated with LPS for 3 d (left panel). The frequencies of IRF4^hiPAX5^lo and IRF4^loPAX5^hi cells were plotted (right panel). Data are representative of three independent experiments. **(E)** Western blot analysis of FOXO1, IRF4, BLIMP1, PAX5, and BACH2 expression in purified splenic B cells from tamoxifen-treated mice of each genotype unstimulated or stimulated with LPS for 3 d. Data are representative of three independent experiments. **(F)** Purified splenic B cells from tamoxifen-treated mice of each genotype (n = 3) were stimulated with LPS for 3 d. The amount of IgM in cell culture supernatants was determined by ELISA. **(G)** Sera were collected from tamoxifen-treated naïve mice of each genotype (n = 10, age 8–12 wk). The amount of serum IgM was determined by ELISA. Two-tailed t tests were used to test statistical significance for (D, F, G). Symbols represent individual mice studied. Error bars represent mean ± SEM. ***P < 0.001.

advantage of newly generated stage-specific conditional mouse models to show that AKT1/2 play fundamental, B cell–intrinsic roles in the GC response by mediating the proliferative and differentiation signals provided by two key receptors: the BCR and CD40. This role is specific to AKT1/2 because mice deficient in AKT1/3 have

normal GC response. AKT proteins are thought to be highly redundant because of their high degree of homology. Such functional redundancy has been described for AKT1 and AKT2 during T cell and B cell development (Juntilla et al, 2007; Mao et al, 2007; Calamito et al, 2010). However, evidence that reflects the specific functional

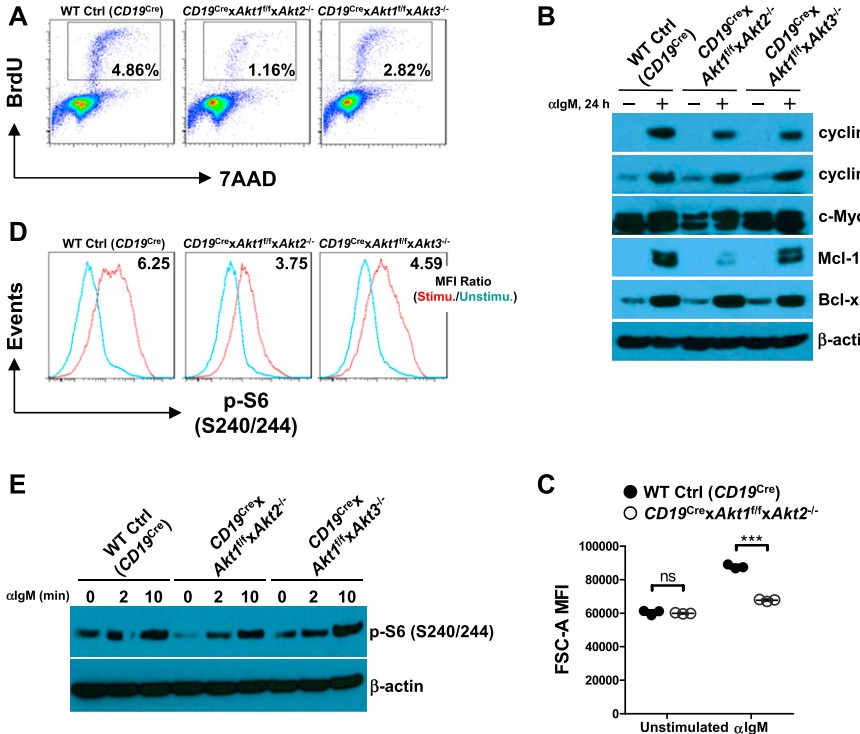

**Figure 6. AKT1/2 deficiency inhibits BCR-activated B-cell survival, growth, and proliferation.**
**(A)** Purified splenic B cells from mice of each genotype were stimulated with anti-IgM for 24 h. The cells were pulsed with BrdU for 1 h before harvest, fixed, and permeabilized, followed by staining with anti-BrdU and 7-AAD. The percentages of cells in the S phase are shown. Data are representative of three independent experiments. **(B)** Western blot analysis of cyclin D2, cyclin D3, c-Myc, Mcl-1, and Bcl-xL expression in purified splenic B cells from mice of each genotype unstimulated or stimulated with anti-IgM for 24 h. Data are representative of two independent experiments. **(C)** Cell size (forward scatter area [FSC-A]) of splenic B cells from mice of each genotype (n = 3) unstimulated or stimulated with anti-IgM for 24 h. **(D)** Purified splenic B cells from mice of each genotype were unstimulated or stimulated with anti-IgM for 10 min. The cells were then fixed and analyzed by flow cytometry for p-S6 (S240/244). mTORC1 activity was measured by p-S6 (S240/244) MFI ratio of stimulated versus unstimulated conditions. Data are representative of three independent experiments. **(E)** Western blot analysis of p-S6 (S240/244) in purified splenic B cells from mice of each genotype unstimulated or stimulated with anti-IgM for indicated time. Data are representative of two independent experiments. Two-tailed t tests were used to test statistical significance for (C). Symbols represent individual mice studied. Error bars represent mean ± SEM. ***$P$ < 0.001.

roles fulfilled by individual isoforms of this family in GC response is absent. Here, we identify a previously uncharacterized specific function in controlling the development of the GC.

Our results showed that a major function of AKT1/2 is to promote B cell proliferation by up-regulating p-S6 expression (Fig 6D and E) and two cell cycle regulators, cyclin D2 and c-Myc (Fig 6B). p-S6 is critical for regulating protein synthesis and cell growth which is a prerequisite for cell division. B-cell proliferation plays an essential role in GC formation and affinity maturation. Cell proliferation provides DNA template for SHM and defines the dynamic relationship of B-cell clonal expansion in the DZ and selection in the LZ. Along with their impact on proliferation, AKT1/2 deficiency resulted in a compromised GC response and consequently plasma cell formation (Fig 1), which rendered the mice unable to produce high-affinity antigen-specific antibody (Fig 2).

Although AKT1/2 are important for the survival of mature B cells in part via induction of Mcl-1 expression (Figs 6B and S9) and critical for the maintenance of GC B cells (Fig 1F and G), our results showed that enforced expression of Bcl2 in GC B cells was unable to rescue their loss in AKT1/2-deficient mice (Fig 7A). Therefore, we propose that the deletion of AKT1/2 in GC B cells affects a subset of centrocytes in the LZ that is subjected to positive selection, rather than the survival of GC B-cell population as a whole, based on two observations. First, BCR-mediated and AKT-dependent phosphorylation of FOXO1 is likely a property of the GC LZ (Luo et al, 2018). It is possible that FOXO1 activation may be restricted to a small fraction of LZ cells that undergoes BCR-mediated selection of the hypermutated antigen receptor for improved antigen affinity in the LZ. Second, lack of CD40 or its ligand, or administration of anti-CD40L at any time during the GC reaction, leads to a rapid involution of the GC (Kawabe et al, 1994; Xu et al, 1994;

Han et al, 1995), indicating that CD40 signaling has a critical role in GC response. Importantly, both CD40 and BCR signals are required to synergistically induce c-Myc and p-S6 in GC B cells (Luo et al, 2018). In agreement with these observations, we found that AKT1/2 deficiency significantly inhibited GC formation and affinity maturation (Fig 2). More interestingly, the loss of AKT1/2-deficient GC B cells was rescued by in vivo anti-CD40 administration (Fig 8A), suggesting that CD40 could deliver the most potent rescue signals in GC B cells. Indeed, we observed strong and synergistic effect of the CD40 and BCR on cell proliferation and induction of c-Myc, p-S6, cyclin D3, and Mcl-1 in AKT1/2-deficient B cells (Figs 8B–D, S8, and S9), which coincided with reduced levels of both p-FOXO1 (T24) and p-FOXO1 (S256) likely because of AKT3-mediated FOXO1 phosphorylation and subsequent degradation (Fig S9). As previously shown, FOXOs may cause apoptosis and cell cycle arrest by reducing the expression of Mcl-1, D-type cyclins, and c-Myc, respectively (Schmidt et al, 2002; Obrador-Hevia et al, 2012; Wilhelm et al, 2016), thereby the reduced levels of p-FOXO1 by combined anti-CD40 and anti-BCR stimulation may abrogate the negative effect of FOXO1 on cell survival and proliferation.

The PI3K/AKT pathway is a central mediator of metabolic regulation in immune cells and GC B cells are under immense proliferative stress, so we hypothesized that the loss of AKT1/2-deficient GC B cells could be partially attributed to limited energy and nutrient supply to GC B cells. Indeed, we found that AKT1/2-deficient B cells are smaller with reduced glucose uptake, decreased MM, and lower OCR relative to WT Ctrl (*CD19*^Cre) B cells upon BCR stimulation (Figs S10A and 8E and F). Furthermore, we found that anti-CD40 stimulation significantly restored the reduced MM and OCR in BCR-stimulated AKT/2-deficient B cells to levels comparable with those in BCR-stimulated WT Ctrl (*CD19*^Cre) B cells (Fig 8E and F), which could partially account for the

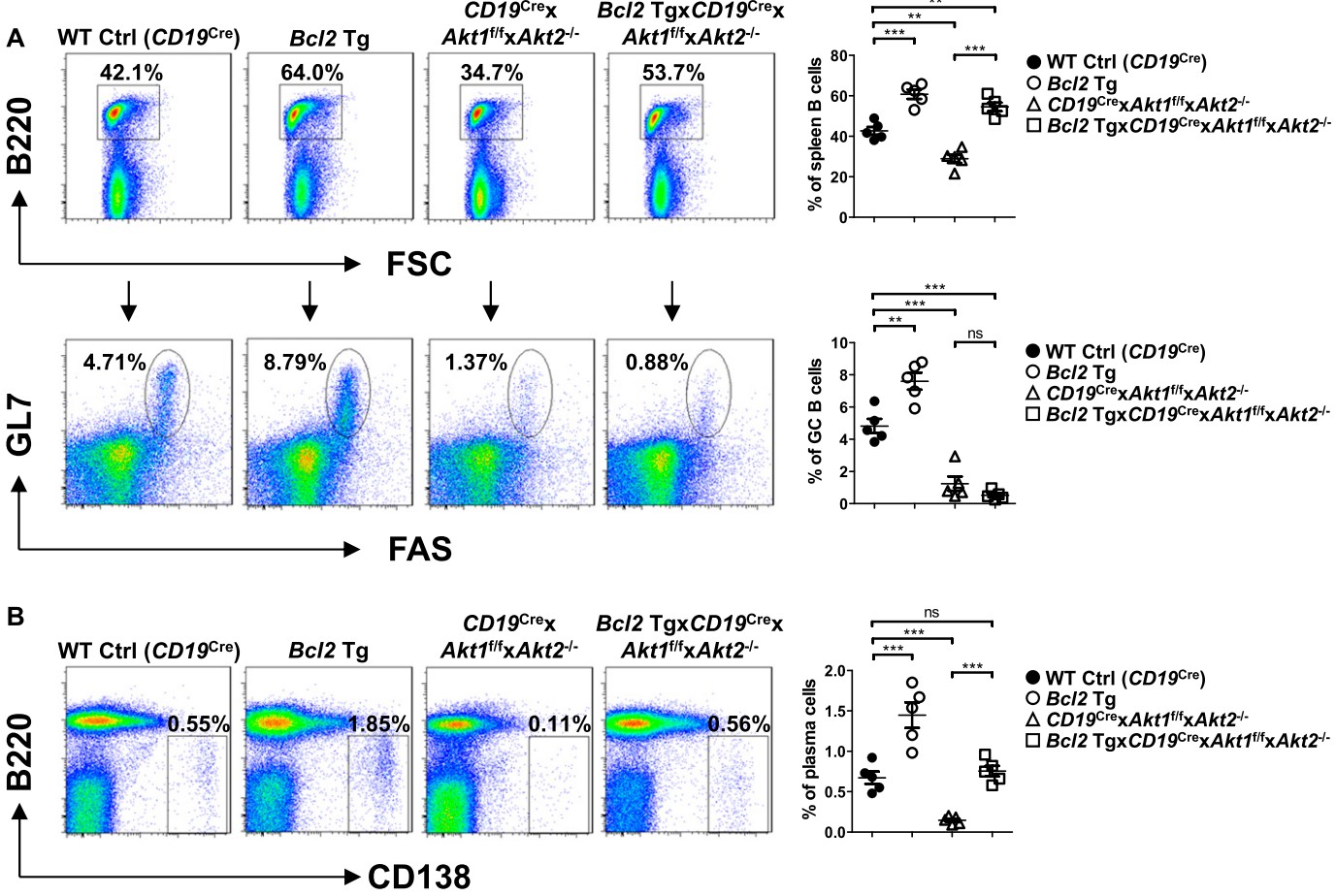

**Figure 7. Enforced Bcl2 expression fails to rescue the loss of AKT1/2-deficient GC B cells in vivo.**
WT Ctrl (*CD19*^Cre), *Bcl2* Tg, *CD19*^Cre × *Akt1*^f/f × *Akt2*^−/−, and *Bcl2* Tg × *CD19*^Cre × *Akt1*^f/f × *Akt2*^−/− mice (n = 5 for each genotype) were immunized with SRBCs on day 0 and analyzed on day 7 postimmunization. **(A)** Representative FACS plots (left panel) and the percentage of total B220⁺ B cells (right panel, top) and GL7⁺FAS⁺ GC B cells as a percentage of total B220⁺ B cells (right panel, bottom) in the spleen were assessed by flow cytometry. **(B)** Representative FACS plots (left panel) and the percentage of B220^lo CD138⁺ plasma cells in the lymphocytes gate (right panel) in the spleen were assessed by flow cytometry. One-way ANOVA was used to test statistical significance for (A, B). Symbols represent individual mice studied. Error bars represent mean ± SEM. **P < 0.01; ***P < 0.001.

rescue of the loss of AKT1/2-deficient GC B cells by in vivo anti-CD40 administration. Together, these results indicate that AKT1/2 are primarily required for B cells to cope with the increasing energy demands encountered upon B-cell activation and would, thus, license the B cells for cyclic reentry to undergo consecutive rounds of hypermutation and selection at the LZ stage after stimulation of the CD40 and BCR.

Effective GC physiology requires controlled expression of FOXO1 by PI3K/AKT signaling. We found that AKT1 predominantly controlled plasma cell differentiation via FOXO1 inactivation (Fig 5C). It is consistent with the notion that high PI3K/AKT activity suppresses FOXO1 function to promote rapid production of plasma cells secreting mainly IgM (Omori et al, 2006). The increased plasma cell generation in FOXO1-deficient B cells is most likely due to the up-regulation of IRF4 expression upon induction of plasmablast differentiation, in the presence of normal Blimp1 up-regulation and PAX5 down-regulation (Fig 5D and E). How IRF4 regulates plasma cell differentiation is unclear. Considering that IRF4 promotes CD8⁺ effector T cells and Th1 cell differentiation via anabolic metabolism, including enhanced glucose uptake, aerobic glycolysis, and

mitochondrial activity (Man et al, 2013; Kratchmarov et al, 2017), we hypothesize that IRF4 could play an analogous role in plasma cell induction. This hypothesis needs to be tested in the future. Collectively, our results provide evidence for a major role of AKT-FOXO1 axis in facilitating plasma cell generation by regulating IRF4 expression.

Elevated PI3K signaling is a hallmark in human cancers with substantial oncogenic strength delivered through AKT (Fruman et al, 2017). Isoform-selective inhibitors of PI3K have the potential to specifically target neoplasms based on tissue of origin and limit disruption of normal homeostatic PI3K signaling. Thus, knowledge on functional AKT isoforms in GC B cells can yield important evidence for the rational design of therapeutics in lymphoma. Also, downstream molecules regulated by AKT may have differential roles among various tissues. Notably, FOXO1 has been implicated in the pathogenesis of GC-derived lymphomas. In diffuse large B-cell lymphoma, FOXO1 mutations occur in almost 10% of cases and are associated with decreased survival in patients (Trinh et al, 2013). Most mutations abolish negative regulation by AKT, including the T24A missense mutation, which is frequently present in these patients. In murine

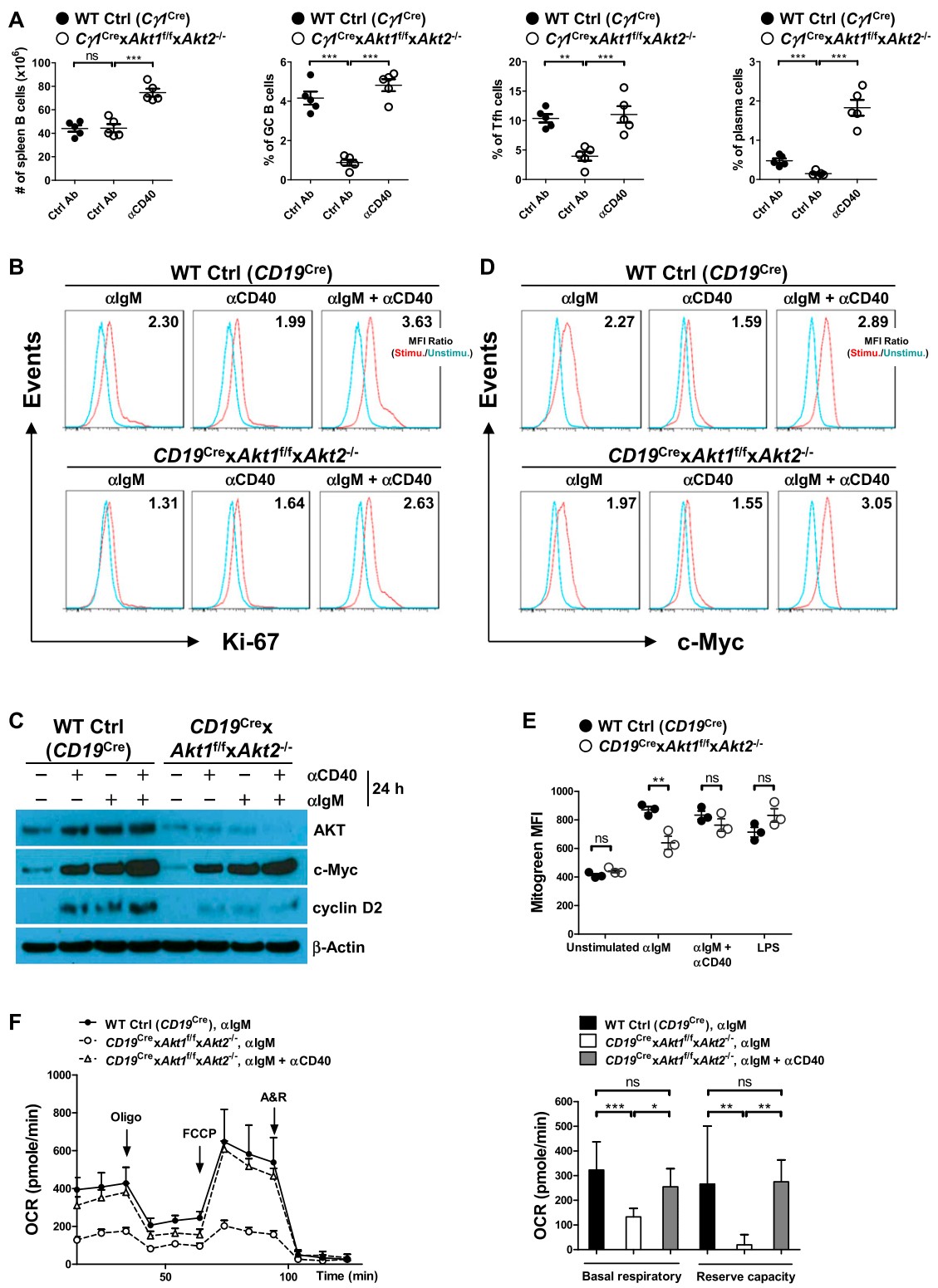

**Figure 8. Anti-CD40 treatment rescues the loss of AKT1/2-deficient GC B cells in vivo.**
**(A)** Mice (n = 5) were i.p.-immunized with SRBCs and i.v.-injected with anti-CD40 and control antibody on day 0. On day 7 after immunization, the B cell number, the frequencies of GC B cells (B220⁺GL7⁺FAS⁺), Tfh cells (CD4⁺CD44⁺CXCR5⁺PD1⁺), and plasma cells (B220^lo CD138⁺) in the spleen were analyzed by flow cytometry. **(B)** Purified splenic B cells from mice of each genotype were unstimulated or stimulated as indicated for 24 h. Cell proliferation was analyzed by Ki-67 intracellular staining. Ki-67 MFI ratios of stimulated versus unstimulated conditions are shown. Data are representative of two independent experiments. **(C)** Splenic B cells from mice of each genotype were purified and cultured with indicated stimuli for 24 h, harvested, and examined by Western blot for AKT, c-Myc, and cyclin D2 expression. Data are representative of

Burkitt lymphoma, $Foxo1^{T24A}$ promotes the proliferation and survival of these cells (Kabrani et al, 2018). Mechanistically, $Foxo1^{T24A}$ largely restricted p-FOXO1 (S256) in the nucleus and robustly induced AID in IgG1-swithced B cells (Fig 4). It is unclear how FOXO1 is involved in the GC B cell lymphomagenesis and how pressure driven by PI3K/AKT signaling contributes to this process. In addition to the role of FOXO1 on GC B-cell proliferation, the downstream induction of AID by FOXO1 may increase aberrant SHM, potentially accumulating more mutations manifesting in aggressive diseases. These findings suggest that FOXO1 activation is pivotal in the etiology of a subset of B-cell lymphomas.

## Materials and Methods

### Mice

$Akt1^{f/f}$ (Cho et al, 2001b), $Akt2^{-/-}$ (Dummler et al, 2006), and $Akt3^{-/-}$ (Tschopp et al, 2005) were kindly provided by Dr Morris J Birnbaum (University of Pennsylvania, USA). $Bcl2$ transgenic mice ($E\mu$-$bcl$-$2$-$22$) (#002319) and $Foxo1^{f/f}$ (#024756) mice were purchased from The Jackson Laboratory. $C\gamma1^{Cre}$, $hCD20$-$Tam^{Cre}$, and $CD19^{Cre}$ mice have been previously described (Rickert et al, 1997; Casola et al, 2006; Khalil et al, 2012). Heterozygous Cre recombinase mice were used in experiments. $Foxo1^{T24A}$ mice were generated by Cyagen Biosciences. All mice were either backcrossed for at least 10 generations to C57BL/6 or maintained on this background. The mice were bred and housed under specific pathogen-free environment in the animal facility at the Sanford Burnham Prebys (SBP) Medical Discovery Institute, and all animal experiments were performed under the regulations of the Institutional Animal Care and User Committee (Supplemental Data 1).

### Flow cytometry analysis

Mouse splenic B cells were purified by negative selection using CD43-specific beads, followed by separation on magnetic-activated cell sorting columns (Miltenyi Biotec). B-cell purity was >98% as measured by FACS analysis using anti-B220 antibody (Thermo Fisher Scientific).

For surface staining, the cells were blocked with anti-CD16/32 (clone 2.4G2; BD Biosciences) and stained with the indicated combination of fluorochrome-conjugated antibodies for 30 min on ice. The following antibodies were obtained from Thermo Fisher Scientific: anti-CD45R (B220) (clone: RA3-6B2), -CD4 (clone: RM4-5), -CD44 (clone: IM7), -IgD (clone: 11-26), -IgM (clone: II/41), -GL7 (clone: GL-7 (GL7)), -Gr-1 (clone: RB6-8C5), -PD1 (clone: J43). Anti-CD138 (clone: 281-2), -CD38 (clone: 90/CD38), -FAS (clone: Jo2), -IgG1 (clone: A85.1), and -IgG3 (clone: R40-82) were purchased from BD Biosciences. Anti-CXCR5 (clone: L138D7) was purchased from BioLegend.

For intracellular staining, the cells were fixed and permeabilized using BD Cytofix/Cytoperm Buffer (BD Biosciences) for 15 min at room temperature. The cells were subsequently washed with 1× BD Perm/Wash buffer (BD Biosciences) before staining with fluorochrome-conjugated antibodies. The following antibodies were obtained from Thermo Fisher Scientific: anti-IRF4 (clone: 3E4) and -Ki-67 (clone: SolA15). Anti-c-Myc (clone: D84C12), -PAX5 (D19F8), and -p-S6 (S240/244) (clone: D68F8) were purchased from Cell Signaling Technology.

All cells were acquired on a FACSCanto flow cytometer using the FACSDiva software (BD Biosciences) and data were analyzed using FlowJo software (Tree Star).

### In vitro plasma cell differentiation and CSR assays

CD43-depleted naive B cells were labeled with eFluor 670 (Thermo Fisher Scientific) according to the manufacturer's protocol and seeded at a final concentration of $0.2 \times 10^6$ cells/ml. For plasma cell differentiation, the B cells were stimulated with 5 $\mu$g/ml LPS (Sigma-Aldrich) for 72 h. For IgG1 or IgG3 CSR, the B cells were stimulated with 5 $\mu$g/ml anti-CD40 agonistic antibody (clone: 1C10; Thermo Fisher Scientific) and 2.5 ng/ml recombinant mouse IL-4 (Thermo Fisher Scientific) or 5 $\mu$g/ml LPS (Sigma-Aldrich) for 96 h, respectively. All B cells were cultured in complete Roswell Park Memorial Institute 1640 (Cellgro; Corning) supplemented with 10% FBS (Sigma-Aldrich), 1× penicillin/streptomycin (Cellgro; Corning), 2 mM GlutaGro (Cellgro; Corning), 1× MEM nonessential amino acids (Cellgro; Corning), 1 mM sodium pyruvate (Thermo Fisher Scientific), and 50 mM $\beta$-mercaptoethanol (Thermo Fisher Scientific).

### In vitro cell cycle assays

For cell cycle analysis, CD43-depleted naive B cells were stimulated with 10 $\mu$g/ml anti-IgM F(ab')$_2$ (Jackson Immuno-Research) for 24 h and then pulsed with 10 mM of BrdU (Thermo Fisher Scientific) during the last hour of incubation before harvest. The cells were then fixed and permeabilized by BD Cytofix/Cytoperm Buffer. After incubation with DNase for 1 h at 37°C, the cells were stained with FITC-conjugated anti-BrdU monoclonal antibody for 30 min at room temperature. 7-aminoactinomycin (7-AAD) was added to each sample right before flow cytometry analysis according to the manufacturer's instructions (BD Pharmingen FITC BrdU Flow Kits).

### In vitro–induced GC B-cell differentiation

CD43-depleted naive B cells were cultured on the feeder cell line CD40LB, in complete Roswell Park Memorial Institute 1640 supplemented with recombinant mouse IL-4 (Thermo Fisher Scientific). On day 5, the cells were harvested and analyzed by flow cytometry.

### In vivo CD40 stimulation

Mice were i.p.-immunized with 200 $\mu$l SRBCs (Colorado Serum Company) and then i.v.-injected with 30 $\mu$g functional-grade purified anti-CD40

---

two independent experiments. **(D)** Flow cytometry analysis of c-Myc expression in purified splenic B cells from mice of each genotype unstimulated or stimulated as indicated for 24 h. c-Myc MFI ratios of stimulated versus unstimulated conditions are shown. Data are representative of three independent experiments. **(E)** MM was measured by MitoTracker Green labeling of purified splenic B cells from mice of each genotype (n = 3) unstimulated or stimulated as indicated for 24 h. **(F)** The OCR of purified splenic B cells from mice of each genotype (n = 3) stimulated as indicated for 24 h were recorded (left panel). Basal respiratory and reserve capacity for stimulated B cells are shown (right panel). Two-tailed $t$ tests were used to test statistical significance for (E). One-way ANOVA was used to test statistical significance for (A, F). Symbols represent individual mice studied. Error bars represent mean ± SEM. *$P < 0.05$; **$P < 0.01$; ***$P < 0.001$.

(clone: 1C10; Thermo Fisher Scientific) or corresponding rat IgG2a isotype control (Thermo Fisher Scientific) on day 0. On day 7 post-immunization, the mice were euthanized, and GC response in the spleen was analyzed by flow cytometry.

## Immunizations

To study GC formation, mice were immunized with 200 µl SRBCs. On day 7 postimmunization, the mice were euthanized and analyzed for the presence of GCs and plasma cells in the spleen by flow cytometry and histology. Titers of SRBC-specific IgM and IgG1 were measured in sera collected on day 7 postimmunization, using a flow cytometry–based method as described previously (McAllister et al, 2017). To study GC maintenance after acute AKT1/2 deletion, SRBC-immunized mice were orally gavaged with tamoxifen for three consecutive days and euthanized as indicated in the respective figure legends. To study affinity maturation, mice were i.p.-immunized with 50 µg alum-precipitated NP25-CGG (Biosearch Technologies). Serum was collected on day 7, day 14, and day 21 after immunization. Antibody titers were determined by ELISA. NP4-BSA– and NP23-BSA–coated plates were used to detect high-affinity antibodies and total titers, respectively.

## Histology

Spleen tissue samples were embedded in Tissue-TEK OCT compound (Sakura Finetek) and frozen at –80°C. Frozen tissue blocks were sectioned, mounted on Superfrost Plus slides (Thermo Fisher Scientific), fixed in ice-cold acetone, and blocked with PBE buffer (PBS containing 1% BSA, 1 mM EDTA, and 0.15 mM NaN3, pH 7.6) with 1% horse serum for 20 min to reduce nonspecific staining. The sections were stained with the following reagents: peanut agglutinin (Vector Labs) and B220 (RA3-6B2; Thermo Fisher Scientific) for 2 h at room temperature. After staining, the sections were washed three times with PBS + 0.5% Tween. Imaging was acquired on a Zeiss Axio ImagerM1 microscope using the Slidebook software (Intelligent Imaging Innovations).

## Subcellular fractionation and Western blot analysis

For analysis of subcellular location of FOXO1 and AKT, CD43-depleted naive *Foxo1*^WT/WT and *Foxo1*^T24A/T24A splenic B cells were unstimulated or stimulated with 10 µg/ml anti-IgM F(ab')₂ (Jackson Immuno-Research) or 5 µg/ml anti-CD40 (clone: 1C10; Thermo Fisher Scientific). Nuclear and cytoplasmic fractions were prepared using the NE-PER Nuclear and Cytoplasmic Extraction reagents (Thermo Fisher Scientific) as per the manufacturer's instruction. For analysis of protein expression in whole cells, whole cell lysates were prepared by direct lysing with radio immunoprecipitation assay buffer plus complete protease inhibitor mixture (Roche).

Protein concentration of subcellular fractionation and whole cell lysates was quantitated using the BCA protein assay kit (Pierce). The samples were boiled in NuPAGE LDS Sample Buffer (Invitrogen) supplemented with β-mercaptoethanol and then were separated by 4–12% polyacrylamide Bis-Tris gel (Bio-Rad) and transferred onto polyvinylidene difluoride membrane (EMD Millipore). The membrane was probed for the indicated proteins. The following antibodies were purchased from Cell Signaling Technology: anti-total AKT (cat. no. 9272), -β-actin (clone: 13E5), -Bcl-xL (clone: 54H6), -BLIMP1 (clone: C14A4), -cyclin D2 (cat. no. 2924), -cyclin D3 (clone: DCS22), -Erk1/2 (cat. no. 9102), -FOXO1 (clone: C29H4), -c-Myc (clone: D84C12), -IRF4 (cat. no. 4964), -PAX5 (clone: D19F8), -p-AKT (S473) (clone: D9E), -p-Erk1/2 (T202/Y204) (clone: D13.14.4E), -p-FOXO1 (S256) (cat. no. #9461), -p-FOXO1 (T24A) (cat. no. 9272), -γH2A.X (S139) (clone: 20E3), -p-S6 (S240/244) (clone: D68F8). Anti-BACH2 (cat. no. 600-401-H35S) and anti-Mcl-1 (cat. no. 600-401-394S) were purchased from Rockland Immuno-chemicals. Anti-p53 (clone: FL-393) from Santa Cruz, anti-AID (clone: 1AID-2E11) from Invitrogen, anti-Tubulin from GeneTex (cat. no. GTX628802). Primary antibodies were then detected with HRP-labeled donkey antirabbit or antimouse antibodies (Jackson Immuno-Research). HRP antibody target proteins were detected by incubating with Supersignal West Pico Chemiluminescent Substrate (Thermo Fisher Scientific) and exposing the blot to x-ray film. Protein bands were quantitated with Image J software (The National Institutes of Health).

## Measurement of MM

MM was determined by Mitotracker Green (Thermo Fisher Scientific) labeling of B cells. Briefly, B cells were unstimulated or stimulated as indicated in the respective figures. The cells were then washed, suspended with PBS, and incubated with 1 µM MitoTracker Green at 37°C for 30 min. The cells were washed once with PBS and subjected to flow cytometry analysis.

## Measurement of cellular OCR

OCR was measured with an XF96 Extracellular Flux Analyzer (Agilent Technologies) and the Seahorse XF Cell Mito Stress Test Kit (Agilent Technologies). The day before assay, the sensor cartridge was hydrated according to the protocol provided by the manufacturer. Before seeding cells, Seahorse XF96 cell culture microplates were coated with poly-D-lysine (50 µg/ml, 100 µl) for 1 h to allow the cells to attach. $1 \times 10^5$ B cells per well were plated the day before analysis and stimulated with anti-IgM F(ab')₂ antibody (10 µg/ml; Jackson Immuno-Research) and anti-CD40 antibody (5 µg/ml; clone: 1C10; Thermo Fisher Scientific), as indicated in the figures. The next day, the cells were washed with warm PBS and incubated with fresh assay media. The cartridge was loaded with three metabolic inhibitors, which were sequentially added to the plate according to the settings of the assay program: oligomycin (an inhibitor of mitochondrial respiratory complex V, 2 µM), followed by FCCP (an uncoupler of mitochondrial oxidative phosphorylation, 0.375 µM), followed by the combination of rotenone (an inhibitor of mitochondria complex I, 1 µM), and anti-mycin A (an inhibitor of mitochondria complex III, 1 µM). The compounds were serially injected to measure basal respiration, coupling efficiency, and reserve capacity. Experiments were performed at the Sanford Burnham Prebys Cancer Metabolism Core.

## Statistical analysis

Results are expressed as mean ± SEM and were determined with one-way ANOVA with the post hoc Tukey's multiple comparisons test and the *t* test for unpaired samples. Statistical significance is

indicated by asterisks (*$P$ < 0.05; **$P$ < 0.01; and **$P$ < 0.001), with $P$ values > 0.05 considered nonsignificant.

## Supplementary Information

## Acknowledgements

We thank the Sanford Burnham Prebys (SBP) vivarium staff for animal care; the SBP Cancer Metabolism Core for help with Seahorse experiments; Dr Morris J Birnbaum (University of Pennsylvania, USA) for providing $Akt1^{f/f}$, $Akt2^{-/-}$, and $Akt3^{-/-}$ mice; Dr Mark Shlomchik (University of Pittsburgh, USA) for providing $hCD20\text{-}Tam^{Cre}$ mice; Dr Daisuke Kitamura (Tokyo University of Science, Japan) for providing CD40LB cell line; and Dr Troy D Randall (University of Alabama at Birmingham, USA) for providing 4-Hydroxy-3-nitrophenylacetyl (NP)-Phycoerythrin (PE) (NP-PE). We are very grateful to all members of our laboratories for the numerous valuable discussions. This work was supported by grant from the National Institutes of Health (R01AI041649 to RC Rickert).

### Author Contributions

Z Zhu: conceptualization, data curation, formal analysis, validation, investigation, methodology, and writing—original draft, review, and editing.
A Shukla: investigation and writing—review and editing.
P Ramezani-Rad: writing—review and editing.
JR Apgar: writing—review and editing.
RC Rickert: conceptualization, resources, supervision, funding acquisition, project administration, and writing—review and editing.

### Conflict of Interest Statement

The authors declare that they have no conflict of interest.

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
