## [Reviewer comments · Life Science Alliance]

Life Science Alliance

The AKT isoforms 1 and 2 drive B cell fate decisions during the germinal center response

Zilu Zhu, Ashima Shukla, Parham Ramezani-Rad, John Apgar, and Robert Rickert

DOI: <https://doi.org/10.26508/lsa.201900506>

Corresponding author(s): Zilu Zhu, Sanford Burnham Prebys Medical Discovery Institute

Review Timeline:	Submission Date:	2019-07-29
	Editorial Decision:	2019-08-20
	Revision Received:	2019-10-28
	Editorial Decision:	2019-11-13
	Revision Received:	2019-11-17
	Accepted:	2019-11-18

Scientific Editor: Andrea Leibfried

Transaction Report:

August 20, 2019

Re: Life Science Alliance manuscript #LSA-2019-00506-T

Dr. Zilu Zhu
Sanford Burnham Prebys Medical Discovery Institute
10901 N. Torrey Pines Rd.
La Jolla, California 92037

Dear Dr. Zhu,

Thank you for submitting your manuscript entitled "The AKT isoforms 1 and 2 drive B cell fate decisions during the germinal center response" to Life Science Alliance. The manuscript was assessed by expert reviewers, whose comments are appended to this letter.

As you will see, the reviewers appreciate your analyses, but think that your conclusions are currently not sufficiently supported. Given the reviewer input received, we would like to invite you to submit a revised version of your manuscript. Some criticisms raised by the reviewers can get addressed by responding to them / discussion, but experimental revision work is also needed. Importantly, better support for your conclusions, e.g., for the timing of GC reduction (rev#1) and for the rescue experiment (both reviewers) is needed. Alternative explanations need to get ruled out (such as cell death underlying the phenotype observed; rev#1) and the results on Foxo1 need clarification (rev#2).

Thank you for this interesting contribution to Life Science Alliance. We are looking forward to receiving your revised manuscript.

Sincerely,

B. MANUSCRIPT ORGANIZATION AND FORMATTING:

Reviewer #2 (Comments to the Authors (Required)):

The manuscript entitled "The AKT isoforms 1 and 2 drive B cell fate decisions during the germinal center response" shows that mice lacking AKT1/2 isoforms in B cells failed to maintain GCs,

consequently mice lack antibody titres (probably because they are not producing antibody-forming cells) and reduced affinity maturation. Data show AKT1/2 isoforms are needed to induce proliferation and survival of B cells after BCR stimulation. Data suggest that the mechanisms behind these findings rely on FOXO1 inactivation promoting IRF4-driven PC differentiation. Lastly, CD40 stimulation rescue B cells by restoring proliferative expansion and energy production.

Introduction

Although introduction nicely summarised supporting literature, this reviewer thinks introduction can be shortened.

Results

B Cell-intrinsic AKT1/2 function is required for GC formation, maintenance, and plasma cell differentiation

Authors discussed GC reduction is mainly on LZ phase of differentiation. An experiment to support this statement is to look for DZ/LZ GC B cells. If authors' assumption is true, DZ GC B cells will be normal while LZ GC B cells will be reduced.

Another possibility is that GC B cells are dying as a consequence of AKT1/2 deletion. Then, they are not able to maintain their proliferation or even mature the affinity. Because there are not cells surviving after AKT1/2 deletion, it would be expected affinity maturation will be dampened. Can authors show the *in vivo* viability/proliferation of GC B cells perhaps by means of active caspase-3 staining/EdU incorporation or intracellular staining with Ki67? That would show if cells are more prone to die after AKT1/2 deletion.

AKT1/2 deficiency impairs high affinity antibody production *in vivo*

It has been proposed memory B cells appear early after GC onset, e.g. day 7 after immunisation with NP-CGG. Have you evaluated earlier time points to see if these differences exist or if something that is happening late during the GC reaction?

Affinity maturation increases with time, did you evaluate later time points e.g. d21? Affinity reduction correlates with the absence of GC B cells, the actual antibody-producing cells won't be produced if GC B cells are absent. If titres are quite low, it is rather difficult to calculate the ratio NP4/NP23. Can you please show the curves for titres calculation (possibly on supplementary figures)?

Authors conclude "these findings demonstrate that AKT1/2 are required for antigen-specific IgG1 affinity maturation". Although affinity maturation seems reduced, the direct link between AKT1/2 deletion and affinity maturation mechanisms is still missing. How AKT1/2 deficiency is interfering with the SHM? Could authors speculate about this?

If AKT1/2 absence is inducing GC B cells death, then reduced affinity maturation is not a direct effect of AKT deficiency. Have the authors assessed the production of NP-specific plasma cells?

AKT1 is the predominant regulator of CSR *in vitro* and *in vivo*

Can author show the summary of data on figure 3B?

CD40 stimulation can rescue loss of Akt1/2-deficient GC B cells *in vivo*

Actually, all B cells were rescued with *in vivo* anti-CD40 administration (Fig 8A.). Why is this happening? If this is an effect on all B cells, then it is not exclusive for GC B cells rescue.

An experiment that might show the effect on rescuing GC B cells exclusively could be *in vivo* anti-CD40 administration on the Cg1CrexAkt1f/fxAkt2^{-/-}, where only GC B cells are affected.

Additional issues

Fig 1. Although it is described on the Results section, it could be easier for the reader, if authors specify in the figure and/or figure legend what control means.

1D. Could you please mention in the figure legend, how many events are in the displayed dot plot? It seems the frequency of cells is really few on the Cy1crexAktf/fxAkt2^{-/-}.

Fig 3C. Have you analysed frequencies of IgG1-producing cells by FACS?

Fig 4B. Which were the criteria for gating in these dot plots? Was it a FMO? There is a clear IgG1+ population in each dot plot but voltages seem different per condition, or probably samples behave

differently under distinct conditions and this is matter of increased auto-fluorescence of the cells? Are all experiments consistently showing the same pattern? This reviewer would take the really clear rounded high IgG1 population as positive and the conclusion would be the same.

Reviewer #3 (Comments to the Authors (Required)):

The PI3K/Akt/Foxo1 pathway is a prominent signaling hub in B cells and is particularly relevant during immune responses. It is also altered in B cell lymphomas by different types of genetic alterations, and thus of great interest. In this manuscript, Zhu and colleagues dissect the role of different isoforms of the Akt kinase in mature B cells and particularly germinal center responses, combining different conditional alleles of Akt 1, 3 and Foxo1. They also include some partial data on the impact of FOXO1 T24A mutations in these responses, which occurs in a subset of germinal center-derived B cell non-Hodgkin lymphomas. The authors find that signaling through Akt1/2 is essential to mount successful germinal center responses. They show, in ex vivo and in vivo studies, that the activity of these enzymes is necessary for germinal center formation, immunoglobulin class switch recombination and affinity maturation of antibodies. They also find that these enzymes are important in the signaling cascade determining plasma cell differentiation. Most of these effects are due to defects in B cell proliferation, survival and metabolic fitness upon B cell receptor stimulation. Interestingly, many of these defects can be rescued (in the context of germinal center B cells) by co-stimulation through the CD40 receptor, a signal also essential for germinal center formation and homeostasis. The authors provide some results explaining how CD40 is able to correct these defects at the molecular level.

There is significant interest in better understanding the roles of the PI3K/Akt/FOXO1 pathway in germinal center biology, given its essentiality to the biology of this compartment and its suspected relevance during lymphomagenesis. Thus, the results reported in this manuscript can be of interest to a wide range of readers. The added description of a couple of new, original mouse models can also be of interest to many.

However, a few points would need to be addressed experimentally to help better interpret some of the key findings in the manuscript. Addressing these points would also add conceptual clarity to some of the points discussed in the manuscript.

>Main Points:

1-Mentions to "control" mice in all figures and throughout the manuscript is a bit confusing. For some experiments, Cgamma1-Cre only mice are used as controls, while in others the authors use Cg1-Cre x Akt2-/- mice. This seems important since, based on Fig. EV1-A, Cg1-Cre Akt2-/- mice show statistically significant defects in, for example, their ability to generate high affinity antibodies. A more homogeneous set of controls or, at least, clear specification of the exact genotype of the animals used as controls in each experiment both in the figures and figure legends should be provided (it is not the case, now).

2-Figure 4: Some observations in the behavior of Foxo1 wt and T24A proteins are unexpected and may require some argumentation.

Concurrent phosphorylation of Foxo1 T24 and S256 is expected to recruit 14-3-3 proteins and result in Foxo1 nuclear export. The authors show that this seems to be observed as p-T24 + p-S256 Foxo1 is depleted from nuclear fractions at 30 minutes post-stimulation, although no changes in the cytosolic fraction or total Foxo1 are observed (the authors imply degradation of cytosolic foxo1). However, similar oscillations in Foxo1 levels in the nuclear fraction occur with the

T24A/T24A mutant. Note that the levels at 30 minutes are clearly reduced when compared to those at 10 minutes post-stimulation, and the dynamics of total Foxo1 protein are not different from those in wildtype B cells. These results are confusing, at least as how they relate to the authors' interpretation.

3-One important finding in this study is that CD40 activation can rescue key deficiencies of the response of Akt1/2 ko B cells to BCR stimulation. CD40 activation also rescues in significant degree the defects in GC formation, including Tfh numbers. In GC B cells, mTOR activity is dependent on mTOR activation (Ersching et al, Immunity 2017; 46(6)). This activity is essential to support the accumulation of biomass in affinity selected B cells. Notably, the authors show in Fig.6 that an important defect in Akt1/2 ko B cells is a severe reduction in the levels of pS6 in response to BCR stimulation. Thus one could reason that the ability of CD40 to rescue the defects of Akt1/2 ko B cells would be due to a "restoration" of mTOR activity. However, pS6 levels have not been analyzed in the experiments shown in Fig.8, despite the severe reduction in pS6 levels in Akt1/2 ko B cells (Fig.6). This appears to be an important piece of missing evidence.

4-Cyclin D2 is a MYC transcriptional target, and its induction appears to be dependent on the activation of PI3K/Akt signaling and FOXO inactivation (Bouchard et al, EMBO J 2004; 23(14)). Thus, the inability of Akt1/2 ko B cells to activate Cyclin D2 expression, regardless of stimulation and despite the increase in MYC levels, doesn't seem surprising. CD40 stimulation does not seem to rescue these defects, despite upon addition of CD40, Akt1/2 ko B cells are able to divide. The authors suggest that rescue is dependent on Cyclin D3, but no evidence is shown. Are the levels of Cyclin D3 protein or RNA in Akt1/2 B cells changing in response to CD40 stimulation?

5-What occurs to FOXO in these experiments (Fig.8)?

6-In the discussion, the authors mention unpublished results describing enhanced proliferation of Foxo1 T24A B cells in response to CD40 and BCR stimulation. Although this may be consistent with previous studies on the potential role of these mutations in supporting B cell survival and proliferation (Trihn et al, 2013; Kabrani et al, 2018), do not seem to fit with the phenotypes of Akt1/2 ko B cells (which are expected to have constitutively active, non-phosphorylated nuclear Foxo1 but instead, fail to proliferate in response to BCR stimulation). These observations are difficult to interpret in light of the findings reported in this manuscript.

>Additional points:

7-The immunoblot results in Fig. EV2-A suggest that loss of Akt2 (maybe also Akt3) may impact the overall levels of Akt1. Only one sample per genotype is included in this analysis, but this may be important, since Akt2^{-/-} mice seem to actually have a significant defect in antibody affinity responses (Fig. EV1)

8-Figure 4B: Two distinct populations of IgG1⁺ cells are observed in the CSR ex vivo experiments. Does this correspond to differences in the balance between surface and intracellular IgG1 protein?

9-Fig. 4D: The correlation that the authors imply between AID levels and the fraction of cells undergoing CSR does not seem to correspond to the experimental results. Specifically, the levels of AID protein in Foxo1 KO B cells are higher than those in WT or WT/T24A cells (yet, Foxo1 KO cells do not switch); and the levels in WT/T24A cells are not different from those in WT/WT cells, yet they show clear differences in the percentage of IgG1 positive cells. How is this explained?

10-Fig. 5B: The histogram depicting the number of cell divisions in LPS-stimulated B cells would suggest that all B cells in this experiment divided synchronously. Commonly, entry of B cells into cycle upon any cytokine stimulation is asynchronous. Are these results reproducible?

11-Fig. 5D: How many biological replicates for this experiment?

12-In Fig.6B the authors also show that Akt1/2 ko B cells also fail to induce expression of Mcl-1 in response to BCR activation. Loss of mTORC1 signaling and loss of Mcl-1 expression have been previously connected and suggest that Mcl-1 expression is sensitive to metabolic states (Coloff et al, Cancer Res 2011; 71(15) and Mills et al, PNAS 2008; 105(31)). Does Mcl-1 expression change upon addition of CD40? (As shown in Fig.8, CD40 corrects some of the metabolic changes in Akt1/2 ko cells in response to BCR stimulation).

13-Previous studies indicate that an important role for IRF4 during plasma cell differentiation is the regulation of BLIMP1 expression. IRF4 also controls expression of AID (Sciammas et al, Immunity 2006; 25(5)). In Fig.5D, the authors show that while IRF4 levels are increased in Foxo1 ko B cells after LPS stimulation, this does not correlate with an increase in BLIMP1, despite a higher number of B cells appear to adopt a plasma cell phenotype. Can this be reconciled with those previous studies? How do cells become plasma cells in greater numbers without observing an increase in BLIMP1?

14-Why did the authors use CD19-Cre only B cells as controls in the mitochondrial stress studies (seahorse) depicted in Fig. 8F?

Referee #2:

The manuscript entitled "The AKT isoforms 1 and 2 drive B cell fate decisions during the germinal center response" shows that mice lacking AKT1/2 isoforms in B cells failed to maintain GCs, consequently mice lack antibody titers (probably because they are not producing antibody-forming cells) and reduced affinity maturation. Data show AKT1/2 isoforms are needed to induce proliferation and survival of B cells after BCR stimulation. Data suggest that the mechanisms behind these findings rely on FOXO1 inactivation promoting IRF4-driven PC differentiation. Lastly, CD40 stimulation rescue B cells by restoring proliferative expansion and energy production.

Main Points:**Comment 1:**

Although introduction nicely summarized supporting literature, this reviewer thinks introduction can be shortened.

Answer:

According to your suggestion, we shortened the introduction from 1168 to 872 words.

Comment 2:

Authors discussed GC reduction is mainly on LZ phase of differentiation. An experiment to support this statement is to look for DZ/LZ GC B cells. If authors' assumption is true, DZ GC B cells will be normal while LZ GC B cells will be reduced.

Answer:

Thank you very much for your comments. According to your suggestion, we performed experiments to examine DZ and LZ B cells in SRBC-immunized WT Control ($C\gamma 1^{Cre}$) and $C\gamma 1^{Cre} \times Akt1^{fl/fl} \times Akt2^{-/-}$ mice. We found that LZ GC B cells were reduced to a greater extent than DZ GC B cells in AKT1/2 KO mice which led to a relatively higher DZ/LZ ratio in AKT1/2 KO mice than that in WT Control mice.

**Loss of AKT1/2 leads to altered DZ/LZ ratio.**

Mice of each genotype (n=5) were immunized i.p. with SRBCs and analyzed on D7. **(A)** Representative FACS plots depicting the gating strategy for DZ ($B220^{+}GL7^{+}FAS^{+}CXCR4^{hi}CD86^{lo}$) and LZ ($B220^{+}GL7^{+}FAS^{+}CXCR4^{lo}CD86^{hi}$) GC B cells in

the spleen (left panel). **(B)** The ratio of DZ vs. LZ GC B cells was plotted (right panel). Two-tailed t tests were used to test statistical significance for (B). Symbols represent individual mice studied. Error bars represent mean \pm SEM. ***, $P < 0.001$.

Comment 3:

Another possibility is that GC B cells are dying as a consequence of AKT1/2 deletion. Then, they are not able to maintain their proliferation or even mature the affinity. Because there are not cells surviving after AKT1/2 deletion, it would be expected affinity maturation will be dampened. Can authors show the in vivo viability/proliferation of GC B cells perhaps by means of active caspase-3 staining/EdU incorporation or intracellular staining with Ki67? That would show if cells are more prone to die after AKT1/2 deletion.

Answer:

Thank you very much for your comments. In order to rule out the possibility that cell viability influences GC B cell proliferation or even the affinity in Akt1/2 KO mice, we crossed the *CD19^{Cre} x Akt1^{fl/fl} x Akt2^{-/-}* mice with *Bcl2* transgenic mice. Our results showed that enforced *Bcl2* expression failed to rescue the loss of AKT1/2-deficient GC B cells in vivo, suggesting the loss of AKT1/2-deficient GC B cells is not due to the inability of these cells to transmit pro-survival signals. The data can be found in Fig 7. According to your suggestion, we performed experiments to examine GC B cell proliferation in vivo using BrdU incorporation approach that we had previously developed. (Matthew H. Cato et al, Cyclin D3 is selectively required for proliferative expansion of germinal center B cells. *Mol Cell Biol.* 2011 Jan;31(1):127-37.) We found that AKT1/2 were specifically required for the proliferative expansion of GC B cells in vivo.

AKT1/2 promote GC B cell proliferation in vivo.

Mice of each genotype (n=5) were immunized i.p. with SRBCs. On D7 postimmunization, the animals were injected i.p. with 2 mg of BrdU. At 6 h postinjection, the in vivo proliferation of GC B cells was measured by flow cytometry. **(A)** Representative FACS plots (left panel) and the frequencies of GL7⁺FAS⁺ GC B cells as a percentage of total

B220⁺ B cells (right panel) in the spleen were shown. **(B)** Representative FACS plots for GC B cell proliferation, as indicated by BrdU incorporation (left panel) and the frequencies of B220⁺BrdU⁺ cells in total GC B cells (right panel) were shown. Two-tailed t tests were used to test statistical significance for (A and B). Symbols represent individual mice studied. Error bars represent mean \pm SEM. ***, P < 0.001.

Comment 4:

It has been proposed memory B cells appear early after GC onset, e.g. day 7 after immunization with NP-CGG. Have you evaluated earlier time points to see if these differences exist or if something that is happening late during the GC reaction?

Answer:

Thank you very much for your comments. According to your suggestion, we performed experiments using the approach we previously developed to examine SRBC-specific memory B cells as early as 7 days postimmunization. (Ellen J. McAllister et al, New Methods To Analyze B Cell Immune Responses to Thymus-Dependent Antigen Sheep Red Blood Cells. J Immunol. 2017 Oct 15;199(8):2998-3003.) We found that AKT1/2 deficiency significantly inhibited SRBC-specific memory B cell formation on D7 postimmunization.

Loss of AKT1/2 inhibits SRBC-specific memory B cell formation.

Mice of each genotype (n=5) were immunized i.p. with SRBCs and analyzed on D7 postimmunization. eFluor670-loaded SRBCs were incubated with splenic cells on ice for 20 min before flow cytometric analysis. **(A)** Detection of SRBC-specific B cells. Representative FACS plots (left panel) and the frequencies of SRBC-specific B cells (B220⁺eFluor670⁺) as a percentage of total B220⁺ B cells in the spleen (right panel) were shown. **(B)** Detection of SRBC-specific memory B cells. Representative FACS plots (left panel) and the frequencies of SRBC-specific memory B cells (B220⁺eFluor670⁺GL7⁺CD80⁺) as a percentage of total SRBC-specific B cells in the spleen (right panel) were shown. Two-tailed t tests were used to test statistical

significance for (A and B). Symbols represent individual mice studied. Error bars represent mean \pm SEM. **, $P < 0.01$; ***, $P < 0.001$.

Comment 5:

Affinity maturation increases with time, did you evaluate later time points e.g. d21? Affinity reduction correlates with the absence of GC B cells, the actual antibody-producing cells won't be produced if GC B cells are absent. If titers are quite low, it is rather difficult to calculate the ratio NP4/NP23. Can you please show the curves for titers calculation (possibly on supplementary figures)?

Answer:

Thank you very much for your comments. According to your suggestion, we performed experiments to examine the NP23-specific IgG and IgM tires in the serum of NP25-CGG-immunized WT Control ($C\gamma 1^{Cre}$) and $C\gamma 1^{Cre} \times Akt1^{flf} \times Akt2^{-/-}$ mice on D7, D14, and D21 postimmunization. We found that the titers of serum NP23-specific IgG and IgM were low on D7 postimmunization. The tires increased with time and peaked on D21. Whereas the titers of NP23-specific IgM were comparable between WT Control ($C\gamma 1^{Cre}$) and $C\gamma 1^{Cre} \times Akt1^{flf} \times Akt2^{-/-}$ mice, the titers of NP23-specific IgG were less in $C\gamma 1^{Cre} \times Akt1^{flf} \times Akt2^{-/-}$ mice than those in WT Control ($C\gamma 1^{Cre}$) mice on D14 and D21 postimmunization. We did examine the titers of serum NP-specific IgG and IgM from the mice in Fig 2 on D21 postimmunization. Since the titers of NP-specific IgG slightly increased on D21 compared to those on D14 in $C\gamma 1^{Cre} \times Akt1^{flf} \times Akt2^{-/-}$ mice and the results on D21 were consistent to those on D14, so we just showed the results on D14 as a representative in Fig 2. We are happy to provide the data on D21 if requested.

Loss of AKT1/2 inhibits the production of NP23-specific IgG.

Mice of each genotype (n=9) were immunized i.p. with NP25-CGG and the serum were collected on D7, D14, and D21 postimmunization. The titers of NP23-specific IgG (left panel) and IgM (right panel) were measured by ELISA. Statistical analysis was done with two-way ANOVA. Error bars represent mean \pm SEM. ***, $P < 0.001$.

Comment 6:

Authors conclude "these findings demonstrate that AKT1/2 are required for antigen-specific IgG1 affinity maturation". Although affinity maturation seems reduced, the direct link between AKT1/2 deletion and affinity maturation mechanisms is still missing. How AKT1/2 deficiency is interfering with the SHM? Could authors speculate about this?

Answer:

Thank you very much for your comments. SHM has been considered to be an event at DNA level. Cell proliferation can provide a large amount DNA templates for SHM. Given that a critical role of AKT1/2 in GC B cell proliferation, we speculate that loss of AKT1/2 inhibits SHM-mediated antibody affinity maturation. We mentioned this in the discussion.

Comment 7:

If AKT1/2 absence is inducing GC B cells death, then reduced affinity maturation is not a direct effect of AKT deficiency. Have the authors assessed the production of NP-specific plasma cells?

Answer:

Thank you very much for your comments. We agree that AKT1/2 are important for the survival of GC B cells. Because enforced Bcl2 expression didn't rescue the loss of AKT1/2-deficient GC B cells in vivo, we think that reduced affinity maturation is largely caused by impaired cell proliferation in AKT1/2-deficient GC B cells. In the 2nd and 3rd paragraphs of discussion, we explained the possibilities how cell proliferation affects the antibody affinity maturation and GC positive selection. According to your suggestion, we performed experiments to assess the production of NP-specific plasma cells. We found that the frequencies of NP-specific plasma cells were significantly reduced in AKT1/2 KO mice compared to that in WT Ctrl mice 21 days after NP-CGG immunization.

Loss of AKT1/2 inhibits NP-specific plasma cells formation.

Mice of each genotype (n=5) were immunized i.p. with NP25-CGG and analyzed on D21 postimmunization. **(A)** Representative FACS plots for NP-specific B220^{lo}CD138⁺ plasma cells. Single splenic cells were stained with anti-B220, anti-CD138, and NP-PE (a kind gift from Dr. Troy D. Randall in University of Alabama at Birmingham) for detection of NP-specific plasma cells. **(B)** The frequencies of NP-specific plasma cells (NP⁺B220^{lo}CD138⁺) as a percentage of total NP-specific cells in the spleen were shown. Two-tailed t tests were used to test statistical significance for (B). Symbols represent individual mice studied. Error bars represent mean ± SEM. ***, P < 0.001.

Comment 8:

Can author show the summary of data on figure 3B?

Answer:

According to your suggestion, we summarized the data in Fig 3A and B.

Comment 9:

Actually, all B cells were rescued with in vivo anti-CD40 administration (Fig 8A.). Why is this happening? If this is an effect on all B cells, then it is not exclusive for GC B cells rescue. An experiment that might show the effect on rescuing GC B cells exclusively could be in vivo anti-CD40 administration on the *Cg1CrexAkt1f/fxAkt2^{-/-}*, where only GC B cells are affected.

Answer:

Thank you very much for your comments. We think that the rescue of B cells by in vivo anti-CD40 administration could partly be caused by upregulation of Mcl-1 (an antiapoptotic protein of the Bcl2 family) expression by anti-CD40 stimulation. The data can be found in Fig S8. According to your suggestion, we performed experiments to show the effect on rescuing GC B cells with in vivo anti-CD40 administration in the *Cγ1^{Cre} x Akt1^{flf} x Akt2^{-/-}* mice. Consistent to our previous findings, we found that in vivo anti-CD40 administration significantly rescued the loss of GC B cells in the *Cγ1^{Cre} x Akt1^{flf} x Akt2^{-/-}* mice. We replaced the old data with the new data in Fig 8A.

Additional points:

Comment 10:

Fig 1. Although it is described on the Results section, it could be easier for the reader, if authors specify in the figure and/or figure legend what control means.

Answer:

Thank you very much for your suggestions. We specified "control" mice in manuscripts. For example, if *Cγ1^{Cre}* only mice are used as controls, we specify them as "WT Ctrl (*Cγ1^{Cre}*)" in both figures and figure legends. We also added and mentioned that "Heterozygous Cre recombinase mice were used in experiments" in the Materials and Methods.

Comment 11:

1D. Could you please mention in the figure legend, how many events are in the displayed dot plot? It seems the frequency of cells is really few on the *Cγ1cre x Aktf/f x Akt2^{-/-}*.

Answer:

According to your suggestion, we added the numbers of events to be displayed in the Dot Plot graph in the legends of Fig 1D.

Comment 12:

Fig 3C. Have you analyzed frequencies of IgG1-producing cells by FACS?

Answer:

Yes, we did surface IgG1 staining on the GC B cells in the SRBC-immunized mice by FACS. The results showed that AKT1-deficient mice had more IgG1 expression on GC B cells than AKT2 or AKT3-deficient mice. The results were consistent to the serum SRBC-specific IgG1 production in those mice in Fig 3C. Because of the limited space, we did not show these data. We are happy to provide them if requested.

Comment 13:

Fig 4B. Which were the criteria for gating in these dot plots? Was it a FMO? There is a clear IgG1+ population in each dot plot but voltages seem different per condition, or probably samples behave differently under distinct conditions and this is matter of increased auto-fluorescence of the cells? Are all experiments consistently showing the same pattern? This reviewer would take the really clear rounded high IgG1 population as positive and the conclusion would be the same.

Answer:

Thank you very much for your comments. We used FOXO1 KO B cells as the negative control since it has been shown that the FOXO1 depletion caused a nearly complete block in IgG1 switching. (Jose J. Limon et al, mTOR kinase inhibitors promote antibody class switching via mTORC2 inhibition, PNAS November 25, 2014 111 (47) E5076-E5085). Please note that the cells were not fixed and it was surface staining. Two distinct populations of IgG1+ cells could partly be due to different proliferation rates of B cells stimulated by cytokines because Ig class switching requires cell proliferation. We repeated the experiments under the same conditions (everything is same except the mouse genotype). The results consistently showed the same pattern as previous results. According to your suggestion, we gated the round high IgG1 population as positive. The new results were shown in Fig 4B instead of the old one.

Referee #3

The PI3K/Akt/Foxo1 pathway is a prominent signaling hub in B cells and is particularly relevant during immune responses. It is also altered in B cell lymphomas by different types of genetic alterations, and thus of great interest. In this manuscript, Zhu and colleagues dissect the role of different isoforms of the Akt kinase in mature B cells and particularly germinal center responses, combining different conditional alleles of Akt 1, 3 and Foxo1. They also include some partial data on the impact of FOXO1 T24A mutations in these responses, which occurs in a subset of germinal center-derived B cell non-Hodgkin lymphomas. The authors find that signaling through Akt1/2 is essential to mount successful germinal center responses. They show, in ex vivo and in vivo studies, that the activity of these enzymes is necessary for germinal center formation, immunoglobulin class switch recombination and affinity maturation of antibodies. They also find that these enzymes are important in the signaling cascade determining plasma

cell differentiation. Most of these effects are due to defects in B cell proliferation, survival and metabolic fitness upon B cell receptor stimulation. Interestingly, many of these defects can be rescued (in the context of germinal center B cells) by co-stimulation through the CD40 receptor, a signal also essential for germinal center formation and homeostasis. The authors provide some results explaining how CD40 is able to correct these defects at the molecular level.

There is significant interest in better understanding the roles of the PI3K/Akt/FOXO1 pathway in germinal center biology, given its essentiality to the biology of this compartment and its suspected relevance during lymphomagenesis. Thus, the results reported in this manuscript can be of interest to a wide range of readers. The added description of a couple of new, original mouse models can also be of interest to many. However, a few points would need to be addressed experimentally to help better interpret some of the key findings in the manuscript. Addressing these points would also add conceptual clarity to some of the points discussed in the manuscript.

Main Points:

Comment 1:

Mentions to "control" mice in all figures and throughout the manuscript is a bit confusing. For some experiments, Cgamma1-Cre only mice are used as controls, while in others the authors use Cg1-Cre x Akt2^{-/-} mice. This seems important since, based on Fig EV1-A, Cg1-Cre Akt2^{-/-} mice show statistically significant defects in, for example, their ability to generate high affinity antibodies. A more homogeneous set of controls or, at least, clear specification of the exact genotype of the animals used as controls in each experiment both in the figures and figure legends should be provided (it is not the case, now).

Answer:

Thank you very much for your suggestions. We specified "control" mice in manuscripts. For example, if Cγ1^{Cre} only mice are used as controls, we specify them as "WT Ctrl (Cγ1^{Cre})" in both figures and figure legends. We also added and mentioned that "Heterozygous Cre recombinase mice were used in experiments" in the Materials and Methods.

Comment 2:

Figure 4: Some observations in the behavior of Foxo1 wt and T24A proteins are unexpected and may require some argumentation. Concurrent phosphorylation of Foxo1 T24 and S256 is expected to recruit 14-3-3 proteins and result in Foxo1 nuclear export. The authors show that this seems to be observed as p-T24 + p-S256 Foxo1 is depleted from nuclear fractions at 30 minutes post-stimulation, although no changes in the cytosolic fraction or total Foxo1 are observed (the authors imply degradation of cytosolic foxo1). However, similar oscillations in Foxo1 levels in the nuclear fraction occur with the T24A/T24A mutant. Note that the levels at 30 minutes are clearly reduced when compared to those at 10 minutes post-stimulation, and the dynamics of total Foxo1 protein are not

different from those in wildtype B cells. These results are confusing, at least as how they relate to the authors' interpretation.

Answer:

Thank you very much for your comments. We think that unchanged total FOXO1 protein levels in Foxo1^{T24A/T24A} mutant B cells could be due to two possible reasons. First, the depletion of p-FOXO1 may not be reflected in total FOXO1 protein because only a small subset of total FOXO1 underwent phosphorylation and degradation upon anti-BCR and anti-CD40 stimulation. This is the case we found in Fig S8. We observed that the total FOXO1 levels were not changed very much although p-FOXO1 (T24) and p-FOXO1 (S256) were largely depleted after anti-BCR and anti-CD40 stimulation for 24 h. Second, the duration of stimulation matters. Short time anti-BCR or anti-CD40 stimulation (30 min) may not be long enough to observe changes in total FOXO1 levels. By contrast, we found that long time anti-CD40 stimulation (96 h) and LPS stimulation (72 h) can significantly decrease total FOXO1 levels (Fig 4D and Fig 5E).

Comment 3:

One important finding in this study is that CD40 activation can rescue key deficiencies of the response of Akt1/2 ko B cells to BCR stimulation. CD40 activation also rescues in significant degree the defects in GC formation, including Tfh numbers. In GC B cells, mTOR activity is dependent on mTOR activation (Ersching et al, Immunity 2017; 46(6)). This activity is essential to support the accumulation of biomass in affinity selected B cells. Notably, the authors show in Fig6 that an important defect in Akt1/2 ko B cells is a severe reduction in the levels of pS6 in response to BCR stimulation. Thus one could reason that the ability of CD40 to rescue the defects of Akt1/2 ko B cells would be due to a "restoration" of mTOR activity. However, pS6 levels have not been analyzed in the experiments shown in Fig8, despite the severe reduction in pS6 levels in Akt1/2 ko B cells (Fig 6). This appears to be an important piece of missing evidence.

Answer:

Thank you very much for your comments. According to your suggestion, we performed experiments to analyze p-S6 (S240/244) levels in the experiments shown in Fig 8. We found that anti-CD40 or anti-BCR stimulation alone poorly induced p-S6 (S240/244) expression in AKT1/2 KO B cells compared to that in WT B cells. Combined anti-CD40 and anti-BCR stimulation strongly and synergistically induced p-S6 (S240/244) in AKT1/2 KO B cells. The new data are shown in Fig S7.

Comment 4:

Cyclin D2 is a MYC transcriptional target, and its induction appears to be dependent on the activation of PI3K/Akt signaling and FOXO inactivation (Bouchard et al, EMBO J 2004; 23(14)). Thus, the inability of Akt1/2 KO B cells to activate Cyclin D2 expression, regardless of stimulation and despite the increase in MYC levels, doesn't seem surprising. CD40 stimulation does not seem to rescue these defects, despite upon addition of CD40, Akt1/2 KO B cells are able to divide. The authors suggest that rescue is dependent on

Cyclin D3, but no evidence is shown. Are the levels of Cyclin D3 protein or RNA in Akt1/2 KO B cells changing in response to CD40 stimulation?

Answer:

Thank you very much for your comments. According to your suggestion, we performed experiments to analyze cyclin D3 levels in the experiments shown in Fig 8. We found that anti-CD40 poorly induced cyclin D3 expression in AKT1/2 KO B cells compared to that in WT B cells. Combined anti-CD40 and anti-BCR stimulation strongly and synergistically induced cyclin D3 in AKT1/2 KO B cells. The new data are shown in Fig S8.

Comment 5:

What occurs to FOXO in these experiments (Fig8)?

Answer:

According to your suggestion, we performed experiments to analyze the levels of total FOXO1, and p-FOXO1 (T24), and p-FOXO1 (S256) in the experiments shown in Fig 8. We found that baseline levels of both p-FOXO1 (T24) and p-FOXO1 (S256) were less in AKT1/2 KO B cells than those in WT B cells. After combined anti-CD40 and anti-BCR stimulation for 24 h, the levels of both p-FOXO1 (T24) and p-FOXO1 (S256) were significantly reduced in both WT and AKT1/2 KO B cells. While the levels of total FOXO1 in AKT1/2 KO B cells were not clearly affected, interestingly, we observed that the levels of total FOXO1 in WT B cells were slightly reduced probably due to protease-mediated degradation indicated by smeared bands. The new data are shown in Fig S8.

Comment 6:

In the discussion, the authors mention unpublished results describing enhanced proliferation of Foxo1 T24A B cells in response to CD40 and BCR stimulation. Although this may be consistent with previous studies on the potential role of these mutations in supporting B cell survival and proliferation (Trihn et al, 2013; Kabrani et al, 2018), do not seem to fit with the phenotypes of Akt1/2 ko B cells (which are expected to have constitutively active, non-phosphorylated nuclear Foxo1 but instead, fail to proliferate in response to BCR stimulation). These observations are difficult to interpret in light of the findings reported in this manuscript.

Answer:

Thank you very much for your comments. We agree that AKT1/2 KO B cells have constitutively active, non-phosphorylated nuclear FOXO1. However, FOXO1 is just one of many downstream targets of the PI3K/AKT signaling pathway. So, it is possible that the phenotypes of AKT1/2 KO B cells are different from those of Foxo1 T24A B cells.

Additional points:

Comment 7:

The immunoblot results in Fig EV2-A suggest that loss of Akt2 (maybe also Akt3) may

impact the overall levels of Akt1. Only one sample per genotype is included in this analysis, but this may be important, since Akt2^{-/-} mice seem to actually have a significant defect in antibody affinity responses (Fig EV1)

Answer:

Thank you very much for your comments. Fig EV2 is now Fig S2 in revised manuscript. We agree that one AKT isoform is possible to compensate for the loss of the other(s) which may reveal the functional interchangeability of AKT isoforms.

Comment 8:

Figure 4B: Two distinct populations of IgG1⁺ cells are observed in the CSR ex vivo experiments. Does this correspond to differences in the balance between surface and intracellular IgG1 protein?

Answer:

Thank you very much for your comments. Please note that the cells were not fixed and it was surface staining. Two distinct populations of IgG1⁺ cells could partly be due to different proliferation rates of B cells stimulated by cytokines because Ig class switching requires cell proliferation. We repeated the experiments under the same conditions, and gated the round high IgG1 population as positive. The negative control is FOXO1 KO B cells since it has been shown that the FOXO1 depletion caused a nearly complete block in IgG1 switching. (Jose J. Limon et al, mTOR kinase inhibitors promote antibody class switching via mTORC2 inhibition, PNAS November 25, 2014 111 (47) E5076-E5085). The new results are shown in Fig 4B instead of the old one.

Comment 9:

Fig 4D: The correlation that the authors imply between AID levels and the fraction of cells undergoing CSR does not seem to correspond to the experimental results. Specifically, the levels of AID protein in Foxo1 KO B cells are higher than those in WT or WT/T24A cells (yet, Foxo1 KO cells do not switch); and the levels in WT/T24A cells are not different from those in WT/WT cells, yet they show clear differences in the percentage of IgG1 positive cells. How is this explained?

Answer:

Thank you very much for your comments. We appreciate the reviewer pointed out that the levels of AID protein in FOXO1 KO B cells are higher than those in WT or WT/T24A. We think that it could be due to an incomplete deletion of FOXO1 by tamoxifen administration. In order to obtain a complete deletion of FOXO1, we increased frequency of tamoxifen administration (50mg/kg) from 3 to 5 consecutive days. We also appreciate the reviewer pointed out that the levels of AID in WT/T24A cells are not different from those in WT/WT cells. Actually, there are barely detectable AID proteins in WT/T24A and WT/WT cells probably due to insufficient protein loading. We increased the amount of protein in Western blot and found that the levels of AID in WT/T24A cells were higher than those in WT/WT cells. The new results are shown in Fig 4D instead of the old one.

Comment 10:

Fig 5B: The histogram depicting the number of cell divisions in LPS-stimulated B cells would suggest that all B cells in this experiment divided synchronously. Commonly, entry of B cells into cycle upon any cytokine stimulation is asynchronous. Are these results reproducible?

Answer:

Thank you very much for your comments. The results are reproducible in 3 independent experiments. We are happy to provide the data if requested. Our data are consistent to previous findings showing that number of cell divisions occurring in response to LPS stimulation for 72 h was largely unchanged between Akt1^{-/-} and Akt2^{-/-} B cells. (Marco Calamito et al, Akt1 and Akt2 promote peripheral B-cell maturation and survival. *Blood*. 2010 May 20;115(20):4043-50).

Comment 11:

Fig 5D: How many biological replicates for this experiment?

Answer:

We did the experiments 3 times (original Fig 5D in manuscripts plus 2 replicates). Please find the data from the replicates in our answer to comment 13.

Comment 12:

In Fig6B the authors also show that Akt1/2 KO B cells also fail to induce expression of Mcl-1 in response to BCR activation. Loss of mTORC1 signaling and loss of Mcl-1 expression have been previously connected and suggest that Mcl-1 expression is sensitive to metabolic states (Coloff et al, *Cancer Res* 2011; 71(15) and Mills et al, *PNAS* 2008; 105(31)). Does Mcl-1 expression change upon addition of CD40? (As shown in Fig8, CD40 corrects some of the metabolic changes in Akt1/2 KO cells in response to BCR stimulation).

Answer:

Thank you very much for your comments. According to your suggestion, we performed experiments to analyze Mcl-1 levels in the experiments shown in Fig 8. We found that baseline expression of Mcl-1 was reduced in AKT1/2 KO B cells. Although anti-CD40 or anti-BCR stimulation alone poorly induced Mcl-1 expression in AKT1/2 KO B cells compared to that in WT B cells, combined anti-CD40 and anti-BCR stimulation strongly and synergistically induced Mcl-1 in AKT1/2 KO B cells. The new data are shown in Fig S8.

Comment 13:

Previous studies indicate that an important role for IRF4 during plasma cell differentiation is the regulation of BLIMP1 expression. IRF4 also controls expression of AID (Sciammas et al, *Immunity* 2006; 25(5)). In Fig5D, the authors show that while IRF4 levels are

increased in Foxo1 KO B cells after LPS stimulation, this does not correlate with an increase in BLIMP1, despite a higher number of B cells appear to adopt a plasma cell phenotype. Can this be reconciled with those previous studies? How do cells become plasma cells in greater numbers without observing an increase in BLIMP1?

Answer:

Thank you very much for your comments. In order to confirm our data in Fig 5D, we independently repeated the experiments twice. Consistent to the results in Fig 5D, we found that IRF4 levels were significantly increased in FOXO1 KO B cells after LPS stimulation (panel A), this did not correlate with a clear increase in BLIMP1 in the two repeated experiments (panel B). Interestingly, previous findings showed that IRF4 deficiency inhibited LPS-induced plasma cell differentiation without affecting Blimp1 expression. (Ulf Klein et al, Transcription factor IRF4 controls plasma cell differentiation and class-switch recombination. *Nat Immunol.* 2006 Jul; 7(7): 773-82, Fig 4a and 4b). Based on these observations, we think that increased IRF4 expression in LPS-stimulated FOXO1 KO B cells may compensate BLIMP1 to promote plasma cell differentiation. We noticed that Klein's data showed that IRF4 was not essential for BLIMP1 expression, whereas Sciammas' data showed that IRF4 was. So, whether or not IRF4 regulates BLIMP1 expression is controversial.

Repeat 1

Repeat 2

Loss of FOXO1 promotes IRF4-driven plasma cell differentiation.

(A) Expressions of IRF4 and PAX5 as determined using flow cytometry of purified splenic B cells from tamoxifen-treated mice of each genotype cultured in LPS for 3 days. **(B)** Corresponding cultures in (A) were analyzed via Western blot for FOXO1, IRF4, BLIMP1, and PAX5 protein.

Comment 14:

Why did the authors use CD19-Cre only B cells as controls in the mitochondrial stress

studies (seahorse) depicted in Fig 8F?

Answer:

We used $CD19^{Cre}$ only B cells as controls in the mitochondrial stress studies (seahorse) depicted in Fig 8F because AKT1/2 KO B cells were obtained from $CD19^{Cre} \times Akt1^{fl/fl} \times Akt2^{-/-}$ mice. So, all mice used in this experiment have heterozygous but uniform $CD19^{Cre}$ genetic background.

November 13, 2019

RE: Life Science Alliance Manuscript #LSA-2019-00506-TR

Dr. Zilu Zhu
Sanford Burnham Prebys Medical Discovery Institute
10901 N. Torrey Pines Rd.
La Jolla, California 92037

Dear Dr. Zhu,

Thank you for submitting your revised manuscript entitled "The AKT isoforms 1 and 2 drive B cell fate decisions during the germinal center response". As you will see, the reviewers appreciate the introduced changes, but a few issues still need to get addressed prior to formal acceptance:

- Please address the remaining concerns of the reviewers
- Please note that we adhere to ICMJE authorship guidelines (author contributions; <http://www.icmje.org/recommendations/browse/roles-and-responsibilities/defining-the-role-of-authors-and-contributors.html>)
- Please incorporate the supplementary material in the main manuscript text - we do not have a length restriction and incorporation will allow for an easier access to all information by the reader
- Please add a scale bar to Fig 1B
- The beta-actin control in Fig. 6E is the same as in fig. S5D. Please clarify whether the experiment for p-ERK and p-S6 has been run at the same time, and if so, add this information to the figure legends, please

A. FINAL FILES:

-- Summary blurb (enter in submission system): A short text summarizing in a single sentence the study (max. 200 characters including spaces). This text is used in conjunction with the titles of

papers, hence should be informative and complementary to the title. It should describe the context and significance of the findings for a general readership; it should be written in the present tense and refer to the work in the third person. Author names should not be mentioned.

B. MANUSCRIPT ORGANIZATION AND FORMATTING:

Sincerely,

Reviewer #2 (Comments to the Authors (Required)):

The manuscript entitled "The AKT isoforms 1 and 2 drive B cell fate decisions during the germinal center response" shows that mice lacking AKT1/2 isoforms in B cells failed to maintain GCs, consequently lack antibody titres and reduced affinity maturation. Data show AKT1/2 isoforms are needed to induce proliferation and survival of B cells after BCR stimulation. Data suggest that the mechanisms behind these findings rely on FOXO1 inactivation promoting IRF4-driven PC differentiation. Lastly, CD40 stimulation rescue B cells by restoring proliferative expansion and energy production.

Authors have answered this reviewer's main concerns.

One more comment, regarding comment 2.

Comment 2:

Authors discussed GC reduction is mainly on LZ phase of differentiation. An experiment to support this statement is to look for DZ/LZ GC B cells. If authors' assumption is true, DZ GC B cells will be normal while LZ GC B cells will be reduced.

Answer:

Thank you very much for your comments. According to your suggestion we performed experiments to examine DZ and LZ B cells in SRBC-immunized WT Control (C β 1Cre) and C β 1Cre x Akt1f/f x Akt2-/- mice. We found that LZ GC B cells were reduced to a greater extent than DZ GC B cells in AKT1/2 KO mice which led to a relatively higher DZ/LZ ratio in AKT1/2 KO mice than that in WT Control mice.

Since these data support authors' statement about LZ GC B cells are the subpopulation mainly affected by the deletion of AKT1/2, it would be important to show these data either in the main figures or in supplemental material.

Then, I agree this paper to be published.

Many thanks for your reply.

Reviewer #3 (Comments to the Authors (Required)):

In this improved, revised version of their manuscript, Zhu et al. provide additional extensive experimental data to convincingly address the main questions presented during peer-review and amend previous problems (e.g. mention to experimental controls).

The results of this manuscript provide important insights to help better understand the role of the PI3K/AKT/FOXO1 axis in germinal center (GC) B cell responses. These insights are particularly important to elucidate the molecular mechanisms that explain the synergy between BCR and T-cell dependent signals during positive selection in GCs. In fact, this pathway has been the focus of several recent studies on basic GC biology, highlighting the considerable interest in the field for this topic.

Just some minor comments/revisions are requested:

1- Since CD40 stimulation synergizes with BCR stimulation to 'normally' induce activation of mTOR, Mcl1 or Ccnd3 (Fig S8), one would guess that such synergy may involve signaling cascades not requiring Akt phosphorylation. It would be helpful to comment on this in the discussion.

2- I would suggest compiling all biological replicates of the experiment shown in Fig5D (IRF4 vs PAX5 dot plots), and summarize it in a graph, to be added as a new panel in this figure.

3- It would be helpful to show the gating strategy and representative flow cytometry plots for the data summarized in panel 4C (IgG1 class switch in vivo); mainly because there was a concern, raised by both reviewers, about the gating of similar phenotypic markers in the ex vivo data.

4- With reference to response to comment #6 in the rebuttal letter: This reviewer acknowledges that there could be additional explanations to the proliferative phenotype of Foxo1 T24A B cells in response to CD40 and BCR stimulation, which the authors mention briefly in the discussion. Given such complexity and the fact that no clear interpretation can be given without additional studies, it would seem more appropriate to leave this point/statement out of the Discussion.

Editor:**Suggestion 1:**

Please address the remaining concerns of the reviewers

Answer:

According to your suggestion, we addressed the remaining concerns of the reviewers. Please see details in our responds below.

Suggestion 2:

Please note that we adhere to ICMJE authorship guidelines (author contributions; <http://www.icmje.org/recommendations/browse/roles-and-responsibilities/defining-the-role-of-authors-and-contributors.html>)

Answer:

Thank you for sending us the ICMJE authorship guidelines. We have carefully read it and understand the policy.

Suggestion 3:

Please incorporate the supplementary material in the main manuscript text - we do not have a length restriction and incorporation will allow for an easier access to all information by the reader

Answer:

According to your suggestion, we incorporated the supplementary material in the main manuscript text.

Suggestion 4:

Please add a scale bar to Fig 1B

Answer:

According to your suggestion, we added scale bars to Fig 1B and showed the information in the legend of Fig 1B in the revised manuscript (lines 976-977).

Suggestion 5:

The beta-actin control in Fig. 6E is the same as in Fig. S5D. Please clarify whether the experiment for p-ERK and p-S6 has been run at the same time, and if so, add this information to the figure legends, please

Answer:

According to your suggestion, we added " The experiments for p-Erk1/2 (T202/Y204) and p-S6 (S240/244) in Figs S6D and 6E were performed at the same time." in the legend of Fig S5D which is now Fig S6D in the revised manuscript (lines 1167-1169).

Reviewer #2 (Comments to the Authors (Required)):

The manuscript entitled "The AKT isoforms 1 and 2 drive B cell fate decisions during the germinal center response" shows that mice lacking AKT1/2 isoforms in B cells failed to maintain GCs, consequently lack antibody titres and reduced affinity maturation. Data show AKT1/2 isoforms are needed to induce proliferation and survival of B cells after BCR stimulation. Data suggest that the mechanisms behind these findings rely on FOXO1 inactivation promoting IRF4-driven PC differentiation. Lastly, CD40 stimulation rescue B cells by restoring proliferative expansion and energy production.

Authors have answered this reviewer's main concerns.

One more comment, regarding comment 2.

Comment 2:

Authors discussed GC reduction is mainly on LZ phase of differentiation. An experiment to support this statement is to look for DZ/LZ GC B cells. If authors' assumption is true, DZ GC B cells will be normal while LZ GC B cells will be reduced.

Answer:

Thank you very much for your comments. According to your suggestion, we performed experiments to examine DZ and LZ B cells in SRBC-immunized WT Control ($C\gamma 1^{Cre}$) and $C\gamma 1^{Cre} \times Akt1^{fl/fl} \times Akt2^{-/-}$ mice. We found that LZ GC B cells were reduced to a greater extent than DZ GC B cells in AKT1/2 KO mice which led to a relatively higher DZ/LZ ratio in AKT1/2 KO mice than that in WT Control mice.

Since these data support authors' statement about LZ GC B cells are the subpopulation mainly affected by the deletion of AKT1/2, it would be important to show these data either in the main figures or in supplemental material.

Then, I agree this paper to be published.

Many thanks for your reply.

Answer:

According to your suggestion, we showed these data in Fig S1 and added the data information in the revised manuscript (lines 126-131 and 1128-1134).

Reviewer #3 (Comments to the Authors (Required)):

In this improved, revised version of their manuscript, Zhu et al. provide additional extensive experimental data to convincingly address the main questions presented during peer-review and amend previous problems (e.g. mention to experimental controls).

The results of this manuscript provide important insights to help better understand the role of the PI3K/AKT/FOXO1 axis in germinal center (GC) B cell responses. These insights are particularly important to elucidate the molecular mechanisms that explain the synergy

between BCR and T-cell dependent signals during positive selection in GCs. In fact, this pathway has been the focus of several recent studies on basic GC biology, highlighting the considerable interest in the field for this topic.

Just some minor comments/revisions are requested:

1- Since CD40 stimulation synergizes with BCR stimulation to 'normally' induce activation of mTOR, Mcl1 or Ccnd3 (Fig S8), one would guess that such synergy may involve signaling cascades not requiring AKT phosphorylation. It would be helpful to comment on this in the discussion.

Answer:

According to your suggestion, we commented on these data in the discussion in the revised manuscript (lines 495-503). Fig S8 is now Fig S9 in the revised manuscript.

2- I would suggest compiling all biological replicates of the experiment shown in Fig 5D (IRF4 vs PAX5 dot plots), and summarize it in a graph, to be added as a new panel in this figure.

Answer:

According to your suggestion, we summarized all biological replicates shown in Fig 5D (IRF4 vs PAX5 dot plots) and showed the data as a new panel in Fig 5D. We also updated the legend of Fig 5D in the revised manuscript (lines 1057-1060).

3- It would be helpful to show the gating strategy and representative flow cytometry plots for the data summarized in panel 4C (IgG1 class switch in vivo); mainly because there was a concern, raised by both reviewers, about the gating of similar phenotypic markers in the ex vivo data.

Answer:

According to your suggestion, we showed the gating strategy and representative flow cytometry plots for the data summarized in Fig 4C (IgG1 class switch in vivo). We also updated the legend of Fig 4C in the revised manuscript (lines 1037-1041).

4- With reference to response to comment #6 in the rebuttal letter: This reviewer acknowledges that there could be additional explanations to the proliferative phenotype of Foxo1 T24A B cells in response to CD40 and BCR stimulation, which the authors mention briefly in the discussion. Given such complexity and the fact that no clear interpretation can be given without additional studies, it would seem more appropriate to leave this point/statement out of the Discussion.

Answer:

We agree with the reviewer and removed this from the discussion.

November 18, 2019

RE: Life Science Alliance Manuscript #LSA-2019-00506-TRR

Dr. Zilu Zhu
Sanford Burnham Prebys Medical Discovery Institute
10901 N. Torrey Pines Rd.
La Jolla, California 92037

Dear Dr. Zhu,

Thank you for submitting your Research Article entitled "The AKT isoforms 1 and 2 drive B cell fate decisions during the germinal center response". I appreciate the introduced changes and it is a pleasure to let you know that your manuscript is now accepted for publication in Life Science Alliance. Congratulations on this interesting work.

DISTRIBUTION OF MATERIALS:

Again, congratulations on a very nice paper. I hope you found the review process to be constructive and are pleased with how the manuscript was handled editorially. We look forward to future exciting submissions from your lab.

Sincerely,
